# Green Copolymers Based on Poly(Lactic Acid)—Short Review

**DOI:** 10.3390/ma14185254

**Published:** 2021-09-13

**Authors:** Konrad Stefaniak, Anna Masek

**Affiliations:** Institute of Polymer and Dye Technology, Faculty of Chemistry, Lodz University of Technology, 90-924 Lodz, Poland; 237771@edu.p.lodz.pl

**Keywords:** polylactic acid, copolymers, catalysts, polymer synthesis, ring-opening polymerization, medical application

## Abstract

Polylactic acid (PLA) is a biodegradable and biocompatible polymer that can be applied in the field of packaging and medicine. Its starting substrate is lactic acid and, on this account, PLA can also be considered an ecological material produced from renewable resources. Apart from several advantages, polylactic acid has drawbacks such as brittleness and relatively high glass transition and melting temperatures. However, copolymerization of PLA with other polymers improves PLA features, and a desirable material marked by preferable physical properties can be obtained. Presenting a detailed overview of the accounts on the PLA copolymerization accomplishments is the innovation of this paper. Scientific findings, examples of copolymers (including branched, star, grafted or block macromolecules), and its applications are discussed. As PLA copolymers can be potentially used in pharmaceutical and biomedical areas, the attention of this article is also placed on the advances present in this field of study. Moreover, the subject of PLA synthesis is described. Three methods are given: azeotropic dehydrative condensation, direct poly-condensation, and ring-opening polymerization (ROP), along with its mechanisms. The applied catalyst also has an impact on the end product and should be adequately selected depending on the intended use of the synthesized PLA. Different ways of using stannous octoate (Sn(Oct)_2_) and examples of the other inorganic and organic catalysts used in PLA synthesis are presented.

## 1. Introduction

Approximately 140 million tons of petroleum-based synthetic polymers are manufactured globally per year, and a considerable amount of them perform in the ecosystem as industrial waste products [1]. Additionally, more and more countries are banning plastic grocery bags, which are said to be responsible for so-called “white pollution” around the world [2]. Even European Union law issues the requirements that concern balanced natural resources management and waste handling [3]. These issues encourage studies on bioplastics that, as biodegradable and ecological replacements, are expected to limit influence on the natural environment [2].

The above-mentioned issues regarding the need for environment protection, decreasing resources of crude oil in the world, and global warming have led to a search for new biodegradable materials. Bio-based plastics have been introduced in order to reduce carbon emissions (because the bio-based raw materials absorb CO_2_ from the atmosphere), and biodegradable plastics have been developed to reduce plastic pollution (because they degrade faster than traditional plastics) [4]. One such material is polylactic acid (PLA). It is made from lactic acid (LA), which is a fermentation product of organic substances such as sweetcorn, rice, soya, potatoes, or whey that is a by-product of the dairy industry. This demonstrates that PLA is produced from renewable resources.

Ecological material has to be environmentally safe in every stage of its “life-cycle”—raw materials sourcing, production, and waste management. Ideal eco-material should be produced from renewable resources, have appropriate mechanical properties, and be easily degradable after exploitation [3]. PLA meets these requirements and, in comparison to other biodegradable plastics, has better mechanical strength, durability, and transparency [4]. PLA is widely applied in food packaging and the medical sector [5]. It is worth noting that several biomedical applications for PLA have been described, e.g., for orthopedic regenerative engineering, as surgical applications in human tissues, or as controlled delivery carriers [6].

In recent years, several studies have been made on the subject of polylactic acid, and new methods of synthesis have been proposed. Several methods of PLA structural modifications have been described [2,4,7,8,9].

This article presents an overview of the studies concerning improving PLA properties and its synthesis. At first, the overall characteristics of PLA is given. The different mechanisms of PLA synthesis and the catalysts used are then presented. Finally, this manuscript concentrates on the variety of PLA copolymers that enable them to change the physical properties of neat PLA. The main goal of this paper is to highlight the fact that, despite PLA being intensely studied, there are still many research aspects that have to be developed, e.g., to improve PLA’s brittleness and gas barrier, by investigating new PLA copolymers. Moreover, attention to the topic of PLA should be drawn because it is a prospective polymer due to its biodegradability and biocompatibility. This review discusses and evaluates recent developments in PLA research with particular reference to its copolymers [6,10,11].

## 2. Lactic Acid (LA)—PLA Monomer

Lactic acid (2-hydroxy propionic acid) (LA) is a starting substrate for obtaining polylactic acid. It is an organic acid that contains asymmetric carbon atoms in its molecules and, as a result, it exists in two different isomers: the levorotatory form called R or D(−) lactic acid and the dextrorotatory form called S or L(+) lactic acid (Figure 1). The signs (−) and (+) signify in which direction a chemical induces a rotation of plane-polarized light. In mammalian organisms, only _L_-LA appears, but if an appropriate strain of bacteria and conditions of process (pH, temperature) are provided, than both isomers or their racemate can be obtained [12,13]. However, the racemic mixture of _D_-LA and _L_-LA (i.e., _D_/_L_-LA) is not recommended for the pharmaceutical, food, and drink industry by reason of the metabolic problems that _D_-LA might cause. It is also not suggested in terms of PLA synthesis because the PLA industry usually needs lactic acid with high optical purity (e.g., ~99% _L_-LA and ~1% _D_-LA) [13].

Lactic acid can be manufactured on an industrial scale, for example, by the catalytic addition reaction of hydrogen cyanide and acetic aldehyde and then the hydrolysis of the obtained cyanohydrin (Figure 2).

There is also the possibility of receiving lactic acid as a result of its fermentation from food industry waste products (potatoes, sweetcorn, sugar beet) and proper bacteria (*Streptococcus*, *Pediococcus*, *Lactobacillus*, *Bifidobacterium*) [14,15,16]. The process, which begins with the hydrolysis of lactose, is presented in Figure 3.

## 3. Polylactic Acid (PLA) Chemical Properties

Polylactic acid (PLA) belongs to the family of aliphatic polyesters (Figure 4). It is also known as polylactide [5].

Homochiral PLA is isotactic and semi-crystalline (up to 60%). Its glass transition is ca. 55 °C and its melting point is ca. 180 °C [5]. Poly(_L_-lactide) (PLLA) is marked by very good tensile strength (60 MPa), small elongation (3–4%), and high modulus (4.8 GPa) [17]. PLA is modified by adding plasticizers such as polyoxyethylene, polycaprolactone, or citrate esters, and in that way PLA’s impact on strength and glass transition is improved [18]. Atactic PLA, made up of heterochiral chains (_L_,_D_), is an amorphous polymer [5].

Saeidlou et al. [19] investigated two types of PLA—semi-crystalline PLA (containing 95% _L_-lactide and 5% _D_-lactide) and amorphous PLA (containing 82% _L_-lactide and 18% _D_-lactide). The results showed that the shear viscosity and the crystallinity increased with increasing _L_-isomer size in the _L_/_D_-isomer mixture. The _L_/_D_-isomer ratio in PLA also impacts on glass transition temperatures, as shown in Figure 5. The higher the content of _L_-stereoisomer is, the higher glass transition temperature occurs.

Scientific research has confirmed that PLA is soluble in both polar and nonpolar solvents, e.g., benzene, tetrahydrofuran (THF), dimethyl sulfoxide (DMSO), acetonitrile, and dioxan [21]. PLA is a completely biodegradable material under industrial composting conditions and assimilable by living organisms [5].

Apart from PLA copolymers, which are precisely described in Section 6, various PLA-based composites have been developed in order to improve PLA properties and applications.

Rodenas-Rochina et al. [22] prepared PLA/hydroxyapatite (HA) composites for application in devices created for bone healing. HA micro- or nanoparticles were dispersed in the polymer matrix.

Natural fibers can be used in order to reinforce the PLA matrix. In a recent study, PLA/flax, PLA/jute, and PLA/falx/jute were fabricated. The concentration of natural fibers in individual composites varied (between 0–50%) by weight. PLA/jute and PLA/flax composites with 40% by weight of fibers improved PLA tensile strength the most. The tensile strength of pure PLA (18.77 MPa) increased to 72 MPa and 45 MPa after flax and jute reinforcement in PLA, respectively [23].

PLA/carbon fiber (PLA/CF) composites are believed to have significant applications in biomedical and engineering sectors. The excellent tensile strength and chemical stability of carbon fibers are the main reasons for interest in the production of PLA/CF composites [24].

## 4. PLA Synthesis—Methods

There are three widely known ways to acquire PLA (Figure 6): lactic acid condensation, lactic acid azeotropic dehydrative condensation, and ring-opening polymerization (ROP) [25]. In this section, every method will be concisely described.

### 4.1. Direct Poly-Condensation

The direct poly-condensation process dehydrates lactic acid into oligomers, which are then further polymerized to PLA with concurrent dehydration to avoid the degradation of polymer molecules by moisture. Nonetheless, removal of water produced from the condensation of lactic acid is noticeably demanding during the final stage of polymerization because the diffusion of moisture in the very much viscous polymeric melt is quite slow. The residual water trapped in the PLA melt can limit the achievable molecular weight and the characteristics of PLA. Therefore, the direct poly-condensation process is little used [27].

### 4.2. Azeotropic Dehydrative Condensation

During the direct LA polycondensation process, there is a difficulty in removing by-produced water. Until 1995, it was believed that a high molecular weight PLA could not be prepared by the polycondensation of LA. Progress was made by Mitsui Chemicals Co. (Tokoy, Japan), because its azeotropic dehydrative polycondensation enabled an increase of the molecular weight of PLA. This method allows the acquisition of PLA with a high molecular weight after a comparatively long reaction time. The dissociated water is removed by means of the so-called azeotropic distillation technique (Figure 7). Solvents with a high boiling point are used for this method [28].

The azeotropic dehydration condensation reaction of LA provides high molecular weight PLA without the use of chain extenders or adjuvants [10]. However, this process has been problematic because of organic solvent usage, which made this method ecologically unattractive [28].

### 4.3. Ring-Opening Polymerization

Ring-opening polymerization (ROP), catalyzed by organometal catalysts, is a method that consists of converting lactide (the cyclic dimer of lactic) to PLA (Figure 8) [29].

Initially, lactic acid is dehydrated and poly-condensed into its oligomers at high temperature and under vacuum in order for moisture to be removed. Lactide is then obtained from catalytic depolymerization of these short PLA chains under reduced pressure. The meso-lactide, lactic acid, and residual moisture can be removed from the optically pure D or L form of lactide by diverse means such as crystallization or distillation. Eventually, the purified lactide is polymerized by a ring-opening polymerization reaction into PLA at temperatures above the melting point of lactide and below the degradation temperatures of PLA. The unreacted lactide (around 5%) must be removed from PLA, and the flowing PLA resin is solidified or/and crystallized into pellets. While the ROP of lactide is conducted, there is little/no moisture to be removed from the molten PLA resin [13].

Three different ROP mechanisms for PLA can be specified: cationic, anionic, and coordinative [14,30]. They differ from each other on the grounds that the lactide ring is opened at different positions depending on the polymerization initiator used. A bond between an oxygen and a carbon belonging to an acyl group of atoms (A) or a bond between an oxygen and a carbon of an alkyl group of atoms (B) can be broken. This is presented in Figure 9 [5].

The cationic ROP mechanism for PLA (Figure 10) consists of breaking the alkyl–oxygen bond of the lactide ring. The propagation mechanism starts with the positively charged lactide ring being opened at the alkyl–oxygen bond by an S_N_2 attack by the trifluoromethanesulfonate (triflate) anion. The triflate end-group then combines with another molecule of lactide again in an S_N_2 manner to obtain a positively charged opened lactide. Polymerization proceeds as the triflate anion again opens the charged lactide [31].

In this mechanism, the possibility of racemization appears because, during every stage of polymer chain propagation, substitution of a monomer occurs in a chiral center. A low molecular weight polymer is yielded with this method. Hence, it is not used on an industrial scale [5].

In the anionic ROP mechanism for PLA (Figure 11), alkali metals alkoxides (e.g., CH_3_OK) act as the initiators of the process. In this process, during the propagation mechanism, the acyl–oxygen bond is cleaved by the initiator’s anion attack. In the obtained PLA macromolecules, an oxygen atom of the alkoxide end-group that propagates has a negative charge, so the configuration of these particles cannot be changed. The PLA synthesis with alkali metal alkoxide as an initiator is marked by good reaction rate, few unwanted side reactions, and high efficiency [5]. It has been shown that using primary alkoxides similar to the aforementioned potassium methoxide enables one to obtain well-defined polymers with insignificant racemization, termination, or transesterification. Racemization was lower than 5% when started with 99.9% pure _L-_lactide [32,33].

In coordinative mechanisms (Figure 12) the initiator combines with lactide after previous acyl-oxygen cleavage. In this type of reaction, alkoxides or carboxylates, including covalent bonds between oxygen and copper atoms or unoccupied p or d orbitals, are used as initiators. Zinc, aluminum, titanium, and stannous alkoxides are the most widely known. In industrial conditions in PLA synthesis, tin(II) 2-ethylhexanoate (also known as stannous octoate) is used as a coordinative initiator [5]. This compound will be more precisely described as a catalyst in the next section of this paper.

## 5. PLA Synthesis—Catalysts

In this section, the topic of the catalysts used in PLA synthesis will be investigated, and several examples will be described.

When PLA synthesis is conducted, an appropriate catalyst is usually used. During this process, catalysts have an impact on the length of the synthesis, and the choice of catalyst should also depend on the desirable application of the manufactured PLA. PLA polymerization reaction time fluctuates between a few minutes and more than 100 h, depending on the catalyst used, the solvent, and the expected molecular weight [9].

A popular catalyst used in PLA synthesis is stannous octoate—Sn(Oct)_2_. It is an effective compound that gives high PLA molecular weights. Molecular weight distributions obtained with Sn(Oct)_2_, compared with other catalysts used in PLA synthesis, are presented in Table 1.

Sn(Oct)_2_ utility results in PLAs with a low degree of racemization (even at high temperatures). Furthermore, it has low toxicity. Sn(Oct)_2_ is a catalyst currently used in aliphatic polyester synthesis that produces atactic PLA chains [37]. It should be noted that Sn(Oct)_2_ is accepted by the FDA (Food and Drug Administration, Silver Spring, MD, USA) for biomedical applications [38]. Figure 13 presents the structure of Sn(Oct)_2_ and the compounds, which are organometallic catalysts and organocatalysts, used in PLA polymerization [6].

### 5.1. Stannous Octoate

Several studies concerning stannous octoate as a catalyst in PLA synthesis have been made. One of them was carried out on the function of polymerization temperature, time, and concentration of catalyst Sn(Oct)_2_. During the polymerization of _L_-lactide, PLA with the highest value of intrinsic viscosity (M_v_ = 10^6^) was synthesized when catalyst concentration was low (0.015 wt %) and at the temperature of 100 °C. A nonionic insertion polymerization mechanism was suggested [39].

Hyon et al. [40] used the same catalyst and obtained PLA with the maximum molecular weight at a catalyst concentration of 0.05% at a temperature of 130 °C. Furthermore, at prolonged polymerization, the decrease in M_v_ and higher polymerization temperature was ascribed to thermal depolymerization of the resultant polylactides.

Other studies have compared the use of tin and zinc compounds. The kinetics and mechanism of _L_-lactide bulk polymerization using stannous octoate and zinc bis (2,2-dimethyl-3,5-heptanedionate-O,O’) was taken into consideration. Up until 80% conversion, the rate of polymerization using Sn(Oct)_2_ was higher than that with zinc-containing catalyst. However, at conversions over 80%, the latter catalyst gave the higher rate of polymerization. The accelerating effect on the polymerization was caused by crystallization of the newly formed polymer. It is suggested that the reason why the differences in the rate of polymerizations at high conversion for the two catalysts is observed is that a difference in crystallinity of the newly formed polymer occurs. Additionally, contaminants in the catalyst and the monomer are thought to be the true initiators. Initiation, as well as polymerization, proceeds through a Lewis acid catalyzed transesterification reaction between an activated lactone and a hydroxyl group [41].

The mechanism of the reaction between the lactide and Sn(Oct)_2_ is recognized as a coordination-insertion mechanism. An alcohol molecule, such as MeOH or the propagating hydrolyzed lactide, which acts as an initiator, exchanges with the octoate ligands. Coordination of lactide to the metal center then occurs. In the next step, the activated nucleophilic alkoxide proceeds by the opening of the lactide. The propagation caused by the generated linear monomer begins, which appears as subsequent lactide coordination and alkoxide insertion until the metal-alkoxide bond is cleaved by termination reactions. Using this method, the PLA obtained includes an ester end group derived from the initiator [42].

### 5.2. Stannous Octoate and Lewis Base

Stannous octoate can lead to relatively rapid lactide polymerization, but it is also known to have a negative effect on the PLA molecular weight and properties. The backbiting and intermolecular transesterification reactions, which occur not only during the lactide polymerization but also during any further melt processing, are responsible for these complications [43]. A solution for this issue has been proposed. The addition of an equimolar amount of a Lewis base, particularly triphenylphosphine on stannous octoate, considerably improves the lactide polymerization rate in bulk. What is more, the kinetic effect has been accounted for by coordination of the Lewis base onto the metal atom of the initiator, making the insertion of the monomer into the metal alkoxide bond of the initiator easier [9].

### 5.3. Stannous Octoate and Distannoxane

The copolymerization of lactide (LA) with mevalonolactone (ML) was performed. Copolymerizations with a monomer ratio LA/ML = 10 were prepared in the presence of two different catalysts: stannous octoate or 1-ethoxy-3-chlorotetrabutyldistannoxane (distannoxane). It was found that, in the presence of Sn(Oct)_2_, branched polymer formation continued by a macromonomer formation step, accompanied by side reactions such as ester exchange and/or alcoholysis. However, copolymerization catalyzed by distannoxane proceeded without side reactions. Due to multiangle laser light scattering and size-exclusion chromatography (MALLS-SEC) analysis and differential scanning calorimetry (DSC), measurement of the formation of branched polymers was also indicated. The following mechanism for the polymerization system using Sn(Oct)_2_ is proposed. Initially, _L_-LA reacts with a pendant hydroxy group of ML; _L_-LA cannot be polymerized using neat stannous octoate, except when in the presence of a protic compound, which is the actual initiator. Side reactions such as the aforementioned ester exchange, alcoholysis, or cyclic oligomer formation can then occur with the extension of the polymerization time. Until _L_-LA is consumed, polymerization and depolymerization (side reaction) are balanced. This can be an explanation of why the polymerization appears to have stopped in the late phase of this reaction stage. Subsequently, the lactone group of monomeric (unreacted) ML and ML units existing at the initiating terminal of a macromer is ring-opened. Therefore, a considerable increase in the molar portion of the ML units in the copolymer is seen at this point. The hydroxy groups of both macromer ML and monomeric can act as initiating groups [44].

### 5.4. Bi(III) Acetate and Creatinine

The catalytic activities of four compounds towards the ROP of _L,L_-lactide were compared: Bi(III) acetate (Bi(OAc)_3_); creatinine, a Sn(Oct)_2_-based system; and a system catalyzed by enzymes. In every case, high and moderate molar mass poly(_L_-lactide)s were obtained. The Bi(OAc)_3_-based system was akin to Sn(Oct)_2_ at 140 °C. The reaction mechanism when using bismuth compounds was recognized as a coordination-insertion mechanism. On the other hand, the reactivity of creatinine (following the coordination mechanism) was lower than that of Bi(OAc)_3_ but highly comparable with that of the enzyme lipase *Pseudomonas fluorescens* [45].

### 5.5. Aluminum Based Catalysts

During the ring-opening polymerization of _D_-lactide, catalyst systems based on aluminum alkoxides are reported to give a polylactide marked by controllable molecular weight with narrow dispersion [25]. It was found that ZnEt_2_ and its complex with aluminum isopropoxide (Al(OiPr)_3_) gave fast polymerization with low transesterification when _D,L_-lactide was polymerized in bulk at 150 °C [46]. When Al(OiPr)_3_ was used in bulk at 100 °C, comparable results were obtained. Moreover, using proton nuclear magnetic resonance (NMR), it was discovered that all chains included isopropoxy ester end-groups and molecular weights correlative to the number of alkoxide groups [47]. This shows that an acyl-oxygen cleavage is a way by which the lactide polymerization is initiated by the aluminum alkoxide. Furthermore, all alkoxide groups of an initiator are active initiating species. It was observed that there was no transesterification at temperatures less than 150 °C, which yields polymers with a narrow molecular weight distribution [47]. The kinetics and the mechanism of the aluminum alkoxide polymerization in solution have been studied [48,49]. One of the findings was that, after an initial induction period, the polymerization is first-order in both monomer and initiator. Despite that, for Al(OiPr)_3_ in toluene at 70 °C, there are three active sites per aluminum molecule, which is contrary to other lactone polymerizations where the number of active sites is less due to catalyst aggregate formation [48,49]. The mechanism involving the insertion of the lactide into the aluminum-alkoxide bond with lactide acyl-oxygen cleavage is presented in Figure 12 as a ROP coordinative mechanism for PLA.

### 5.6. Summary

It has been known for many years that high molecular weight PLA can be obtained by the ring-opening polymerization of lactide. An astonishing number of catalysts that have been used in order to initiate this polymerization has been reported. Nevertheless, there is still a need to devise safer, faster, and more stable catalysts. Considering the development of a single-step extrusion process, there are doubts concerning the safety of the catalysts that are left in the final polymer, which is later used in human body applications [9].

## 6. PLA Copolymers

In order to improve several PLA physical properties, there have been studies concerning PLA modification by creating its copolymers. In this section, such solutions will be described. However, at the outset of this section, nomenclature concerning polymers is presented in Figure 14.

### 6.1. PLGA Copolymer

Poly(_D,__L_-lactide-co-glycolide) (PLGA) is a linear copolymer of lactic acid (LA) and glycolic acid (GA) [4]. Figure 15 shows its chemical structures. PLGA copolymers prepared at different ratios were investigated, and its degradation time was compared. The results shown in Table 2 indicate that the copolymerization of LA with GA shortens PLA degradation time by as much as three-fold, from 6 to 2 months. Furthermore, the higher the content of polyglycolide (PGA) in the copolymer, the shorter the observed degradation time.

There are several ways to obtain PLGA. Solution poly-condensation of LA and GA at a temperature above 120 °C yields low molecular weight PLGA [4]. Using ring-opening polymerization of glycolide and lactide with a metal catalyst (e.g., stannous octoate), high molecular weight PLGA can be prepared. If there is a demand for PLGA with non-possible toxic metallic contaminations, which is favorable for bio-medical applications, enzymatic polymerization should be used [53]. Additionally, it is known that the sequence of PLGA has a significant impact on its degradation rate. Random PLGA degrades quicker than sequenced ones. 4-(dimethylamino) pyridinium p-toluenesulfonate (DPTS) and 1,3 diispropylcarbodiimide (DIC) have been used as catalysts in order to prepare repeating sequence copolymers [58]. Ring-opening polymerization (ROP) and segmer assembly polymerization (SAP) were used. The latter is a method that allows preparation of sequence polymers and gives numerous possibilities for periodic copolymer synthesis. In this approach, sequenced oligomers (segmers) are first prepared and then polymerized. It is a paradigm that shows the convergence of synthetic, organic, and polymer chemistries [59,60,61]. Figure 16 shows details of the previously described issue.

Because of a suitable time for degradation of PLGA to occur, this biocompatible copolymer can be used, e.g., in drug delivery systems. Depending on the needs, choosing an appropriate LA/GA ratio and the polymer’s molecular weight allows the creation of desirable material. Furthermore, the two methods of PLGA synthesis described and the wide variety of available catalysts enable adjustments to suit the needs of the end product.

### 6.2. Metal-Centered Star-Shaped PLA (Co)Polymers

A star polymer is a polymer composed of branched macromolecules that contain only one common branch unit. Hence, in shape, it looks like a star. If the polymer’s arms are chemically different then it is termed a miktoarm star polymer [50]. There are known star-shaped PLA polymers and copolymers with metal ion cores. Fe^2+^, Fe^3+^, Eu^3+^, and Ru^2+^ ions are used as metal centers. Star structures can be produced by the combination of coordination chemistry with controlled or living polymerization. Sn(Oct)_2_ is commonly used as a catalyst [6]. These materials are designed for a specific role, and because of their stimuli-responsiveness, luminescent materials for drug delivery can be conceived [37]. Luminescent ruthenium tris(bipyridine)-centered star block copolymers, consisting of PLA as the hydrophobic core and poly(acrylic acid) (PAA) as the hydrophilic corona, may provide a multifunctional drug delivery system with the capability of optical imaging. Star copolymers were obtained by the consecutive ROP of _D,L_-lactide, atom transfer radical polymerization (ATRP) of tert-butyl acrylate, and finally, the tert-butyl end-groups were hydrolyzed [62].

ATRP—atom transfer radical polymerization—is a particular type of a controlled radical polymerization (CRP). ATRP enables entering monomers and cross-linking agents into a polymer chain in a controlled way. This allows the procurement of nearly equally long polymer chains. Their length is determined by the used monomer and initiator ratio. Applying the ATRP method to polymer synthesis allows adjustment of the structure of these polymers to drug delivery requirements. ATRP is based on a redox reaction between dissolved transition metal ions and alkyl halides in order to regulate an equilibrium between their active species (radicals) and dormant species (Figure 17). The initiation stage is faster than the propagation stage in order to provide the concurrent growth of all polymer chains. In the ATRP system, there are four elementary reagents: initiator (P_n_-X), activator (transition metal complex; Mt^n^/L), deactivator (Mt^n+1^/L), and macroradicals (P_n_*), which grow as a result of adding other monomer particles. Transition metals such as Cu(I), Ni(II), and Fe(II) are the central metals in catalytic complexes. Derivatives of 2,2′-bipyridine, such as N,N,N′,N″,N″-pentamethyldiethylenetriamine (PMDETA), appear as ligands (L). During the reversible redox process, one electron is moved from the transition metal complex (which is in its the lowest oxidation state), and a halogen atom is ripped off a polymer chain (dormant species), which leads to creating a radical and deactivated catalytic complex in a higher oxidation state [63].

Implementing metal ions into polymer structures broadens the possibilities of further research. Customizing the synthesis process in terms of the catalyst used, the provided hydrophilicity/hydrophobicity, and the macromolecule structure, gives a potential opportunity to use the obtained copolymers in areas of nanotechnology, biomedicine, and drug delivery systems.

### 6.3. PEG-PLA Copolymer

PLA and its copolymers can be applied as a base for hydrogel medicinal implants, which can be used as scaffolds for tissue engineering or drug delivery vehicles. For example, the copolymers of lactic acid with poly(ethylene glycol) (PEG) (structure of PEG in Figure 18) form thermo-responsive hydrogels. The physical cross-linking mechanisms of PEG-PLA consist of: ionic or lactic acid segment hydrophobic interactions, chemical bond formation by radical- or photo-cross-linking, and stereocomplexation of _D_- and _L_-lactic acid segments [65].

Eight-arm star block PEG-b-(PLLA)_8_ copolymers, functionalized with pyridine, were used in order to form star block PEG-b-PLLA-py metallo-hydrogels in the presence of transition metal ions (Cu^2+^, Co^2+^, Mn^2+^). These PEG-b-PLLA block the copolymer hydrogels, present distinguished biocompatibility and biodegradability, and could be adopted in a broad range of biomedical and industrial applications [67].

With the use of Michael-type addition reaction, chemically cross-linked PEG-b-PLLA hydrogels were prepared. Eight-arm thiol-terminated star PEG (PEG-(SH)_8_) reacted with acrylated PEG-b-PLLA star block copolymers (PEG-b-(PLLA_12_)_8_-AC) (Figure 19). Excellent mechanical properties presented by these hydrogels were observed. Furthermore, the degradation time of the formed hydrogels oscillated between a few days to several months and was regulated by the incorporated amount of PEG-(SH)_8_. Lysozyme was released from the eminently cross-linked PEG-b-(PLLA_12_)_8_-AC/PEG-(SH)_8_ hydrogels, predominantly by diffusion [68]. Additionally, miktoarm star polymers, PLLA/PEG, were reported to form thermo-responsive hydrogels in water at high concentrations (22.5 wt %) [69].

In order to introduce stable cross-links and thus improve the mechanical properties of the hydrogels formed by PEG-b-PLA star block copolymers, the stereocomplexation, that is co-crystallization, of PLLA and PDLA blocks can also be exploited. Moreover, enantiomeric PEG-PLA star block copolymers with a central PEG core and outer PLA blocks were shown to gelate faster and form stereocomplexed hydrogels with improved mechanical strength, in comparison with triblock PLA-b-PEG-b-PLA copolymers [6].

Regarding degradation issue, four-arm star PEG-b-PLA polymers are thermally stable at biological conditions. Due to relatively short degradation times, PEG-b-PLA star copolymers might be a superb candidate for drug delivery applications. Because, during hydrolytic degradation, the PLA chain length decreased, lactic acid concentration increased. The lactic acid concentration of the medium content PLA samples (PEG/PLA = 2.5/0.8) reached a maximum around day 21 (0.256 mg/mL × 10), suggesting total degradation of the PLA chain. The initial concentration was ca. 0.140 mg/mL × 10. The high content PLA sample (PEG/PLA = 2.5/1.6) at day 21 had an acid concentration of 0.449 mg/mL × 10—the initial concentration was ca. 0.200 mg/mL × 10. Using the aforementioned copolymers as a short-term drug release agent was suggested [70].

The solutions presented in this subsection give information concerning other metal ions that can be used in PLA copolymerization processes as they have key importance in the issue of forming a shape of molecule by creating ligands. On the other hand, the presence of star-shaped PEG-(SH)_8_ is important in terms of degradation and protein release time because of its molecular composition.

### 6.4. PCL-PLA Copolymer

Polycaprolactone (PCL) is a synthetic thermoplastic polymer. It is a linear semicrystalline polyester that is degraded in a natural environment by bacteria and fungi. In order to enhance its properties, copolymers such as poly(glycolide-co-caprolactone) (PGCL) and poly(_L_-lactide-co-ε-caprolactone) (PLCL) have been studied [17].

PCL-PLA copolymers have features of thermoplastic elastomers [71]. A biodegradable PCL-PLA multiblock copolymer was obtained when hexamethylene diisocyanate (HMDI) was added during the chemical reaction. A strong interaction between PCL and PLA was observed. This might indicate that there is a reaction between the PCL hydroxyl, PLA carboxyl, and HMDI isocyanate groups. A copolymer with a mass ratio of PCL:PLA = 80:20 has a Young’s modulus of 2.7 ± 0.7 MPa and tensile elongation at break ca. 790%. This composition is desirable in medicine and technology [72].

A PCL-PLA long-chain branched block copolymer was introduced in order to prepare a biodegradable PLA material with enhanced crystallinity, rheological behavior, and mechanical properties. Adding the PCL-PLA copolymer to the neat PLA improved its tensile toughness without ill effect on the above-mentioned properties. Furthermore, PLA/PCL-PLA blend with 15 wt % of the PCL-PLA copolymer had much better elongation at break (210.7%) than neat PLA (7.1%). The studied copolymer was synthesized in the reaction of single hydroxyl-terminated PLA (PLA-OH) with hydroxyl-terminated three-arm star PCL (PCL-3OH) in the presence of HMDI. HMDI was used as the chain-extending agent [73]. The toughening of PLA, while simultaneously preserving its biodegradability and mechanical properties, should be highlighted. Subsequent studies should focus on seeking the most practical mass ratio between PLA and the PCL-PLA copolymer. The bioplastic material described may find numerous technological applications.

Song et al. [74] studied ABA PCL/PLA/PCL block copolymers and AB PCL/PLA (example structure in Figure 20) block copolymers. The physical and mechanical properties of these compounds are intriguing—especially elongation at break. Elongation at break of the PCL/PLA block copolymer with [CL]/[LA] = 72/28 is 380%, while the value for the PCL/PLA/PCL block copolymer with [CL]/[LA] = 75/25 is 600%, without the sacrifice of tensile strength. The suggestion of using ABA block copolymers as a biomedical material with tough membrane forming properties for a sustained release drug delivery system was presented.

Regarding thermal properties, tubular scaffolds made of poly(_L_-lactide-co-ε-caprolactone) (PLCL) (50:50) random copolymers were synthesized, and DSC analysis showed two potential glass transition temperatures (T_g_) of the scaffolds in the 0 to −40 °C region. No crystalline melting peak was observed, which shows that the PLCL random copolymer is amorphous. The DMA profile indicated two T_g_s: one at −38 °C and a second at −11 °C. This indicated a phase-separated structure of the PLCL studied. PCL and PLA homopolymers had higher T_g_: −60 °C and +50 °C, respectively. Hence, the T_g_ at −38°C aligns with a phase composed of mainly CL units, and the T_g_ at −11 °C signifies the other phase containing more LA moieties. Additionally, the scaffolds had 200% higher elongation at break values than their homopolymers, which indicates the advantages of copolymerizing LA and CL to make flexible and soft copolymer scaffolds. FT-IR analysis confirmed the formation of random copolymers [76].

Elongation at break of PCL-PLA copolymers should be noted. The value of this property is much higher for PCL-PLA copolymers than for PCL and PLA homopolymers. This advantage can be used in the packaging and medical industries.

### 6.5. POSS-PLA Hybrid Copolymer

The physical and mechanical properties of PLA can be greatly improved by developing organic-inorganic hybrid materials. Polyhedral oligomeric silsesquioxane (POSS), consisting of silicon and oxygen atoms arranged in an inner eight-cornered cage with Si atoms positioned at the corners, was synthesized first. The star-shaped POSS-polylactides (POSS-PLAs) with varied PLA arm lengths were then obtained through ring opening polymerization of _D_,_L_-lactide. Eventually, the star-shaped POSS-PLA based polyurethanes (POSS-PLAUs) were formed by cross-linking POSS-PLA and polytetramethylene ether (PTMEG) with HMDI. POSS-PLAUs presented superb shape memory properties. POSS-PLAUs with shorter arm length showed faster recovery speed as a result of the higher POSS core content [77].

The ring-opening polymerization of _L_,_L_-LA, catalyzed by Sn(Oct)_2_ was initiated by the functionalized silsesquioxane cages of the regular octahedral structure. As a result, biodegradable hybrid star-shaped POSS-PLA and linear systems with an octasilsesquioxane cage as a core and PLLA arms were given. Biodegradation of the compounds obtained is assumed on the grounds that both lactide blocks and POSS moieties are biodegradable [78]. Biodegradable POSS lactide systems can be applied in biomedical applications. A unique class of inorganic structures presented by POSS can be utilized in the era of hybrid polymer systems with advantageous properties [79].

In order to reduce the brittleness of PDLLA, a highly branched hybrid copolymer based on polyhedral oligomeric silsesquioxane POSS was composed. POSS-OH was used as the core of the toughening material, and the ring-opening polymerization of ε-caprolactone and _D_,_L_-lactide was then initiated sequentially to create the highly branched POSS-g-poly(ε-caprolactone)-b-poly(_D_,_L_-lactide) (POSS-g-PCL-b-PLA) copolymer with eight PCL-b-PLA arms. Furthermore, POSS-g-PCL-b-PLA/PDLLA nanocomposites were prepared via solution casting. Due to adding the PLA segment, good compatibility and distribution between POSS-g-PCL-b-PLA and the PDLLA matrix were observed. Elongation at break increased, and the yield stress decreased as the POSS-g-PCL-b-PLA content increased. This was due to the core-shell structure of POSS-g-PCL-b-PLA, which considerably improved the toughness of the PDLLA polymer matrix [80].

Synthesis of the star-shaped organic/inorganic hybrid PLLA, based on POSS, was begun from POSS bearing octa(3-hydroxypropyl) moieties [81]. Subsequently, further transformation of POSS-PLA was made. POSS-PLA was changed into the POSS-containing star-shaped organic/inorganic hybrid amphiphilic block copolymers, poly(_L_-lactide)-block-poly(N-isopropylacrylamide) (POSS(PLLA-b-PNIPAM)), by the reversible addition-fragmentation transfer (RAFT) polymerization of N-isopropylacrylamide (NIPAM) (see Figure 21). Star-shaped POSS-PLLA-b-PNIPAM amphiphilic block copolymers self-assembled into vesicles in an aqueous solution. Hydrophilic PNIPAM blocks and the hydrophobic POSS core and PLLA created coronas and the vesicular wall, respectively. The temperature dependence of the hydrodynamic radius (R_h_) for POSS(PLLA_12_–b–PNIPAM_119_)_8_ block copolymers in aqueous solution was investigated with dynamic light scattering (DLS) measurements. When temperature decreased from 34 °C to 30 °C, the R_h_ noticeably increased from 53 nm to 93 nm. This shows that the PNIPAM block in the aggregates is temperature responsive. At temperatures below 30 °C, the R_h_ did not change significantly during the cooling or heating processes, meaning that the phase-transition process of the PNIPAM block is reversible. It can be deduced that, for the cooling process with temperatures below 34 °C, PNIPAM chains began to stretch. The self-assembly morphology of POSS(PLLA–b–PNIPAM) block copolymers was studied by transmission electron microscopy (TEM). Self-assembled vesicular structures of the star-shaped POSS(PLLA–b–PNIPAM) amphiphilic block copolymers in aqueous solution were observed. However, there was a broad disparity in the size of the vesicular aggregates, and the density of the vesicular wall was not uniform. The outer diameter of the vesicles was polydispersed in the range of 20 nm to 35 nm. This size of the vesicles was smaller than the values measured by DLS. This results from the fact that DLS data directly reflects the size of self-assembly aggregates in solution, where the PNIPAM block chains are sufficiently dispersed in water, even although the PNIPAM chains are attached by one end onto the surface of the vesicular wall. The block copolymers described could be exploited in medical and biological fields [81].

The compounds presented in this subsection are distinct from the other examples described in this paper on the grounds that they are inorganic–organic hybrid materials. POSS-g-PCL-b-PLA/PDLLA composite desirably decreases the brittleness of linear PDLLA, which is said to be its one of its biggest drawbacks. Moreover, the amphiphilic and self-assembly character of the POSS(PLLA–b–PNIPAM) block copolymer should be noted. The highlighted features of the aforementioned compounds give POSS-PLA hybrid materials considerable prospects for biomedical applications.

### 6.6. PVA-g-PLA Copolymer

Graft polymers are a subclass of branched polymers. They are composed of blocks connected to the main chain as side-chains, with the chemical constitution of these side-chains differing from those of the main chain [50,51].

The grafting of PLA chains by initiating LA (or LA and GA) polymerization from OH groups of the poly(vinyl alcohol) (PVA) backbone is an example of the “grafting from” method (Figure 22) [82,83]. Such action allows modification of the polymer molecular architecture. This in turn has an impact on the crystallinity and biodegradability of a polymer. Graft polymers can be applied in drug delivery systems.

Wang et al. [83] performed lactide polymerizations in bulk (130 °C) and used stannous octoate as a catalyst. PVA-g-PLA and PVA-g-P(LA-co-GA) copolymers with M_n_ in the range of 75,000–275,000 (SEC) were obtained. They can be applied as drug delivery systems with tunable physicochemical properties such as molecular composition, molecular weight, degree of crystallinity, and both melting and glass transition temperatures that could be adapted to the demands of drug delivery. Release of the hydrophilic dextran was investigated by preparing microspheres. The release profiles depended on degradation characteristics and were further modified by introducing sulfonate groups to poly(vinyl alcohol), which resulted in generating negative charges along the PVA backbone.

Sulfonate modified P(VS-VA)-g-PLGA compounds are especially promising because the degree of sulfonate substitution determines degradation time. One of the samples lost 50% of its mass in 8 days. It seems to be very useful in terms of drug delivery systems. Further research concerning degradation and drug release appears to be necessary. Studies concerning the potential applications of P(VS-VA)-g-PLGA in the packaging industry could also be carried out.

### 6.7. PLA-Glycidol Copolymer

It was found that adding glycidol to PLA chains efficiently led to its branching. Branched polymers were formed in a one-pot approach during polymerization performed in bulk at temperatures above the melting point of LA (100–180 °C); 4–70 ratios of lactide/glycidol (LA/GLY) were used with either BF_3_·Et_2_O as a Lewis acid or diphenyl phosphate as a protic acid. With the use of diphenyl phosphate as a catalyst, a higher level of lactide conversion was observed, and the products were colorless in contrast to BF_3_·Et_2_O usage. The branched structure of the obtained polymers was additionally confirmed with thermal analysis. PLA copolymers were marked by a significant decrease in the melting temperature (T_m_) and melting enthalpy (ΔH_m_) compared with PLA homopolymer. This resulted in a strongly lowered tendency for crystallization. In the copolymer, where [LA]/[GLY] = 17, T_m_ was 130 °C and 134 °C and ΔH_m_ was 4.5 J/kg. By use of the multimodal SEC trace, it is suggested that, at the beginning of the copolymerization, macromolecules of PLA initiated with glycidol molecules were formed, and only later were these macromolecules coupled via terminal ring opening. Initiation with hydroxyl groups occurring simultaneously in macromolecules eventuated in the formation of branches (Figure 23) [84]. It is assumed that a branched structure in the studied system is formed in terms of the activated monomer mechanism of cationic polymerization proceeding in the presence of hydroxyl groups [85].

Besides improving thermal properties, the hydrophilicity of PLA was also markedly enhanced as a result of adding glycidol units, which was confirmed with contact angle measurements. The contact angle of polymer films on a glass plate (with water as a reference liquid) decreased from 95.7° for linear PLA to 62.4° for a copolymer containing 20 mol% glycidol [11]. Better hydrophilicity is important for PLA because degradation increases as material hydrophilicity increases.

Reducing LA-GLY copolymer melting temperature by over 30 °C in comparison with linear PLA might create new possibilities in PLA synthesis, because the temperature of synthesis is one of the key factors. The lower the melting temperature becomes, the weaker are the forces between molecules and, as a result, chemical synthesis is facilitated. Furthermore, carrying out the synthesis at lower temperature is technically more practical. When synthesis is conducted taking melting temperature into consideration, finding the purity of the compound obtained is also simplified. In order to extend the study, other Lewis acids can be tested as catalysts. With regards to the enhanced hydrophilicity of PLA/GLY, copolymer research concerning the degradation properties of these compounds could be desirable in order to analyze the potential biomedical applications of this material.

### 6.8. PLA-Hydroxyoxetanes Copolymers

The bulk copolymerizations of lactide with 4-membered hydroxyetanes (3-hydroxymethyl-3-methyl-oxetane (HMMOX) and 3-ethyl-3-hydroxymethyl-oxetane (EHMOX) were considered) catalyzed by diphenyl phosphate were completed. As a result, branched polymers with molecular weights in the range of 1800–16,600 (SEC measurement with refractive index (RI) detection and against PSt standards) were obtained. Their molecular weights were depended on the lactide/oxetane ratio in the feed. Due to crude reaction mixture SEC traces, it is revealed that a monomodal distribution was executed, in contradistinction to that of lactide/glycidol copolymers. However, just as in lactide/glycidol copolymers, DSC analyses showed a considerable decrease in crystallinity. Investigating copolymer with [LA]/[OX] = 17, T_m_ was 134 °C and ΔH_m_ was 0.5 J/kg. Matrix assisted laser desorption/ionization time of flight (MALDI TOF) analyses carried out for all polymers confirmed the presence of several “branching monomer” units in LA/GLY and LA/OX copolymers [11].

Attempts to increase molecular weights could be made. Additionally, it is worth noting that the authors of studies on LA/OX copolymer reached even lower melting enthalpy than researchers working on the aforementioned LA/GLY compound. Lower melting enthalpy, and hence crystallinity, is essential for PLA compounds because of degradation. Crystalline regions are more resistant to hydrolysis and, as a result, crystalline and semicrystalline polymers are marked by slower degradation rates than amorphous ones [7].

### 6.9. PLA Copolyesters

The “grafting from” method can be used to prepare graft polyesters, where the polymer backbone is also a polyester, by the application of precisely designed comonomers [11].

Using lactide and a specially synthesized monomer, the cyclodepsipeptide cyclo[Glc-Ser(OBz)] (CGS-OBz) copolyesters were prepared. Polymerization was catalyzed by stannous octoate. When polymerizations were performed with 5–15 mol% cyclo[Glc-Ser(OBz)], copolyesters with molecular weights in the range of 20,000–25,000 were obtained. Afterwards, deprotection of hydroxyl groups was conducted, and polylactide chains were grafted onto this polymer, again using Sn(Oct)_2_ as a catalyst of polymerization performed in bulk [86]. The applied strategy is presented in the Figure 24.

In order to scrutinize the crystallinity of PLA-based graft polymers, the DSC analysis method was introduced. Moreover, the copolymers were subjected to hydrolysis with the aim of making a correlation between the polymer architecture and their crystallinity and hydrolytic stability. The result was that the obtained comb-type PLA presented a decrease in crystallinity and an increase in biodegradability compared with linear PLA. Varying the molecular architecture could bring a change in the degradation rate of PLA [86].

Observed correlations between crystallinity, molecular architecture, and the degradability of synthesized copolymers are noticeably detailed, and extending the research on mechanical studies is welcomed. Contact angle measurements would certainly give more data regarding the hydrophilicity of graft comb-type PLA copolymers.

### 6.10. PLA Copolymers—Summary

In summary, the aim of presenting PLA copolymers in this section was to show that PLA copolymerization is a prospective method for changing several PLA properties. Essentially, copolymerizing PLA with compounds such as PGA, PEG, or glycidol shortens PLA degradation time. In the authors’ judgement, this is the main advantage of forming different PLA copolymers. Secondly, creating various PLA copolymers enables the manipulation of PLA hydrophobicity. The ruthenium star block copolymer mentioned has a PLA hydrophobic core and a PAA hydrophilic corona. Additionally, adding glycidol to PLA enhances PLA hydrophilicity, which has an impact on degradation rate and potential biomedical applications (e.g., drug delivery systems). Furthermore, PLA copolymers (e.g., PCL-PLA) are marked by improved toughness and/or elongation at break in comparison with neat PLA. Depending on the needs, diversifying other PLA properties, such as crystallization or contact angle values, makes it possible to compose new PLA copolymers. The fact that both inorganic (e.g., POSS) and organic compounds can be used for PLA copolymerization is worthy of note Many possibilities as regards designing different copolymer architectures also aid this method’s advance. As a result, PLA copolymers with broad applications can be obtained—the next section is focused on this issue.

## 7. PLA Applications

Two main areas of PLA use can be detailed. The first field is represented by products of general use, while the second involves specialistic applications in medicine. The following products made of PLA for general use can be listed: containers, wrapping films, disposable products, and elements of interior furnishings. On the other hand, in medicine, PLA can be applied in the areas of bioresorbent implants, surgical sutures, clamps, clips, surgical masks, or dressings [14,87,88,89,90,91]. PLA copolymers also have special applications.

Studies on the PLA-polyethylene glycol block copolymer (PLA-PEG) and the PLA-p-dioxanone-polyethylene glycol block copolymer (PLA-p-DPEG) have been made. These compounds have been used as carriers for bone morphogenetic proteins (BMPs). BMPs are biologically active molecules able to induce new bone formation, and they are expected to be used clinically in connection with biomaterials, such as bone-graft substitutes to stimulate bone repair. The effect of PLA on the osteoinduction of demineralized bone and the usefulness of PLA as a carrier of BMP were analyzed. It was found that PLA was a satisfactory candidate as a carrier for BMP [92]. Initially, low molecular weight PLA was mixed with BMP in order to form a composite, which was next implanted in the host bone; new bone cells were formed while the degradation of the PLA matrix in the composite was occurring [93]. However, the newly-formed bone was too low in quantity (bone mineral density). For this reason, PLA copolymers with low molecular weight were used to solve these problems [10].

However, in some cases, bone defects occur in positions that require dynamic strength (e.g., the long bone of the leg). In these situations, in order to restore the bone, the BMP/polymer composite has to be combined with a solid biomaterial with good affinity for bone. As an example, a titanium implant featuring a porous surface on which the BMP/polymer composites can be settled is an appropriate material that can be implanted into the bone defect. The new bone formed by the BMP/polymer composites would then firmly surround the titanium implant until the implant and the host bone fused [93]. Thus, even repairing bone defects that require strength would become possible. This process is described in Figure 25 [92]. When the pores of the solid implant are filled with the BMP/polymer composite and implanted, the composite exudes from the pores and forms a layer of bone covering the surface of the biomaterials. This layer of bone may encase the implant and thus improve biological fixation of the biomaterials to the host bone [92].

PLA, in view of its biodegradability, is also used in drug delivery systems (DDS) in which the drug can be released continuously for different periods of time up to one year [10]. PLA and their copolymers, in the form of nano-particles, have been used in several applications in the encapsulation process of many drugs, such as dermatotherapy [94], protein (BSA) [95], hormones [96], restenosis [97], antitumor oridonin [98], and psychotic [99]. These nano-particles were received with different methods, such as solvent displacement, solvent evaporation [100], emulsion solvent diffusion [101], and salting out [99].

A lisinopril-conjugated triblock PLA-PEG-PLA copolymer was synthesized by the reaction of PLA-PEG-PLA copolymer with lisinopril (the antihypertensive drug). Subsequently, the lisinopril-conjugated PLA-PEG-PLA was self-assembled into micelles in an aqueous solution. Conjugated micelles were characterized by a better sustained release profile in comparison with the lisinopril-conjugated copolymer and physically loaded micelles [102]. This micellar formulation of drug conjugated amphiphilic copolymers seems to be a significant achievement in controlled drug delivery.

A PLA-PEG-PLA copolymer was used to prepare microspheres containing paclitaxel. The microspheres obtained had a porous structure inside, which advanced drug release. In vitro release was 49% in one month, making drug action time in the body longer. This drug delivery system avoids inconsistent local concentration on account of drug release. Thus, the therapeutic effect is enhanced [90].

Poly(lactic-co-glycolic) acid nanoparticles were used for loading the drug, paclitaxel. The preparation of nanoparticles was conducted by the emulsion solvent evaporation method in the presence of tocopheryl polyethylene glycol succinate as an emulsion agent. The ability of in vitro drug release of the nanoparticles, encapsulation efficiency, and the drug loading efficiency were investigated. The in vitro release mode of drug-loaded nanoparticles appeared to be two-staged—a fast release in the initial stage and a slower release in a second stage. The encapsulation efficiency was 4.84%, and the drug loading efficiency was 67.35% [103]. In order to enhance the abovementioned values, further studies on the porosity of PLGA nanoparticles could be carried out. Changing the LA/GA mass ratio in a copolymer would have a potential impact on its degradation behavior. The two-staged mode of drug release described can be practically applied in particular medical circumstances.

The combination of PLGA with the antibacterial substance totarol was used as a novel coating solution for surgical sutures. Collected data suggests that the biodegradable suture coating obtained has the potential to reduce the risk of surgical site infections (SSIs)—a nosocomial infection that can result in severe complications after surgical intervention. The coating prevents post-operative biofilm formation during the critical phase of wound healing. It has no negative impact on tissue [104].

Regarding tissue engineering, PCL-PLA copolymer is a relevant compound for this type of applications. Recently, a PCL-PLA copolymer nanofiber has been used in the regeneration of damaged tissue [105]. Moreover, an improvement in the mechanical properties of PCL can be achieved by copolymerization with PLA, enabling its use for orthopedic applications, such as the repair of bone defects [106].

Another copolymer that has the potential to be applied in medicine is PVA-g-PLA. Animal experiments featuring thin copolymer films made of this material were conducted. The thickness of the samples was 0.04–0.06 mm. In comparison with the PLA homopolymer, the copolymer obtained had improved hydrophilicity and flexibility. The films showed a satisfying anti-tissue adhesion effect and applicable degradability in the body of the mouse. The film entirely disappeared after 8 weeks of implantation. The films were also biocompatible, as expected, because no inflammation, hematoma, or infection were noticed [107]. These results are very promising regarding future PVA-g-PLA use for preventing post-operative organ tissue adhesion.

A poly(3,4-ethylenedioxythiophene) (PEDOT) polymer and a biocompatible polymer polylactide were synthesized to design graft copolymers. The weight fraction of 3,4-ethylenedioxythiophene (EDOT) was between 5 and 40%. The compound created can be 3D printed using direct melting extrusion methods; 5:95 wt % PEDOT-g-PLA showed the best performance during the printing process as it improved the ability to control its shape. This compound was used as the patterning material for biocompatibility tests on neonatal cardiac cultures. Tissue-like structures made of cardiomyocytes with fibroblast were developed in PEDOT-g-PLA and cardiomyocyte, improving the approved suitable maturation and functionality of these cells. The promising results of the presented study could assist the progress of generating artificial tissue [108].

## 8. Conclusions

PLA is a biodegradable polymer synthesized from lactic acid. Because PLA is an ecological material, its use could make a positive difference to the worldwide environment.

Regarding PLA synthesis, ring-opening polymerization is a method during which lactide is converted into PLA. The reaction is catalyzed by organometal catalysts. Stannous octoate (Sn(Oct)_2_) is a compound commonly used as a catalyst. It gives PLA weight distributions (M_n_) between 40,000 and 250,000. Sn(Oct)_2_ efficiency can be improved by adding a Lewis base or distannoxane to the synthesis process. Moreover, stannous octoate can be applied in the biomedical sector as it is accepted by the FDA (Food and Drug Administration, Silver Spring, MD, USA). However, studies concerning new catalysts that can be used in PLA synthesis are necessary because there is a known negative impact of Sn(Oct)_2_ on PLA properties. In particular, biocompatible catalysts are desirable.

Apart from the many advantages of PLA, such as appropriate biodegradability, durability, and transparency, it also has some drawbacks: brittleness, relatively high melting temperature, and hydrophobicity. Hence, PLA copolymers are composed with the intent of enhancing specific PLA properties. Diverse compounds in terms of composition (e.g., POSS-PLA hybrid copolymers) and polymer architecture (e.g., PVA-g-PLA or PLA copolyesters) have been created. As shown by DSC analysis, graft PLA copolymers are marked by a lower crystallinity and higher degradation rate than linear PLA. The sequence of individual monomers also has an influence; random PLGAs degrade quicker than sequenced ones. A PLA-glycidol copolymer provides significant improvement in terms of thermal properties and better hydrophilicity. Regarding elongation at break and tensile strength, PCL-PLA copolymers should be highlighted. It can be concluded that PLA copolymers enable the creation of ideal materials that can be applied in certain fields, such as medicine, packaging, or technology. Biocompatible PLA copolymers seem to have considerable potential in the field of biomedical applications such as drug delivery systems or tissue engineering. Studies concerning this subject should be continued.

Polylactic acid might be a crucial material for the plastics industry in the near future due to its environmentally friendly character. Several methods of PLA synthesis and the many possibilities for creating new PLA copolymers make conducting new research desirable. Improving PLA properties and searching for new applications for PLA seem to be the two biggest challenges for the development of PLA-based materials.

## Figures and Tables

**Figure 1 materials-14-05254-f001:**
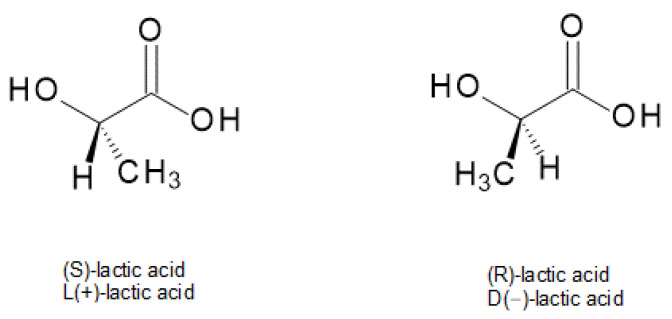
Two lactic acid isomers [13].

**Figure 2 materials-14-05254-f002:**
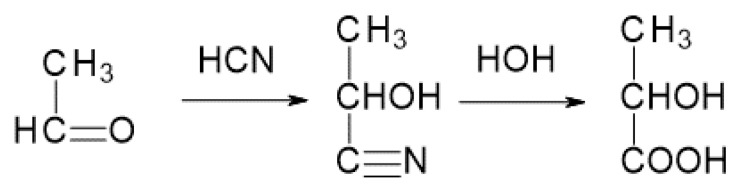
Scheme presenting the method of producing LA from acetic aldehyde using hydrogen cyanide [5].

**Figure 3 materials-14-05254-f003:**
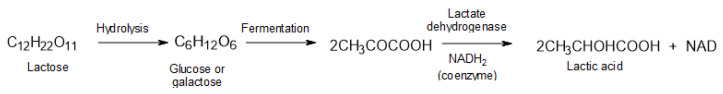
Scheme presenting receiving lactic acid [5].

**Figure 4 materials-14-05254-f004:**
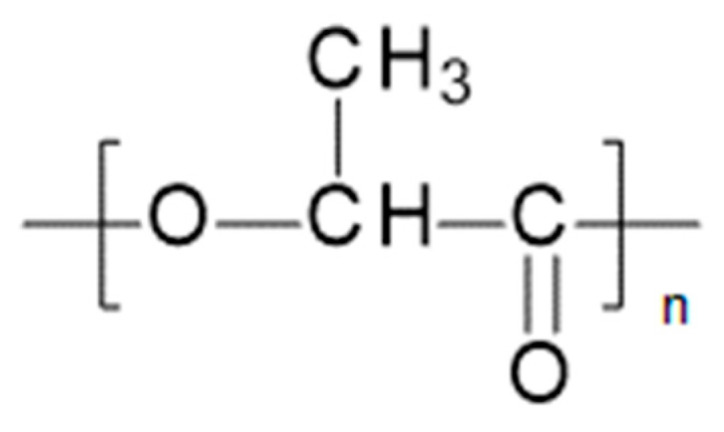
Chemical structure of PLA [5].

**Figure 5 materials-14-05254-f005:**
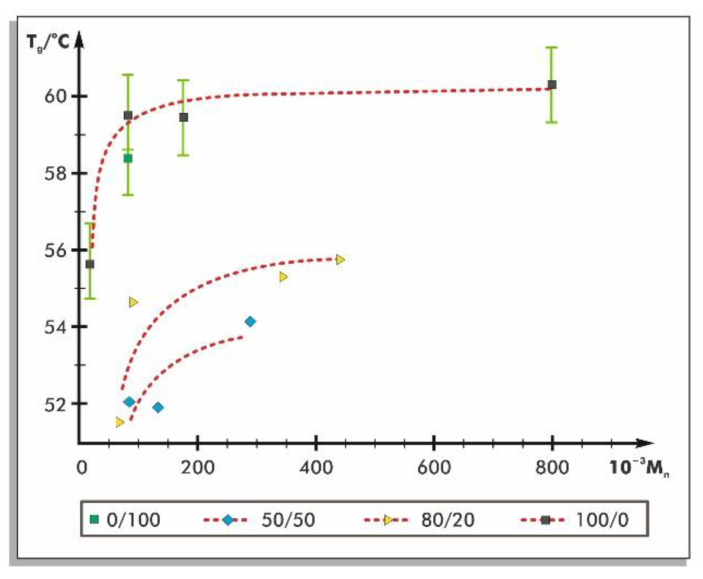
Glass transition temperatures (T_g_) of PLA polymers with different _L_-stereoisomer contents as a function of number-average molecular weights (M_n_). Symbols on the legend represent the _L_/_D_-isomers ratios [20].

**Figure 6 materials-14-05254-f006:**
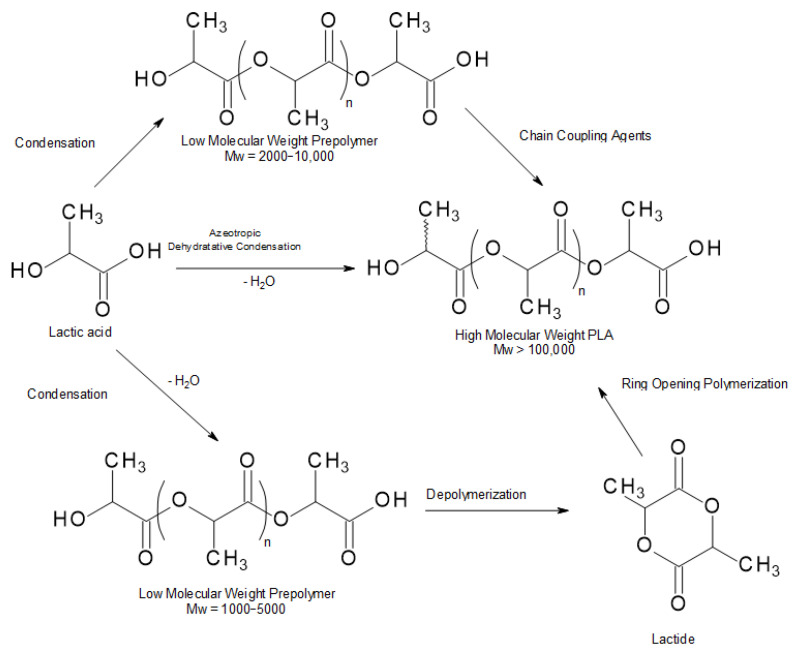
Manufacturing routes to polylactic acid [26].

**Figure 7 materials-14-05254-f007:**
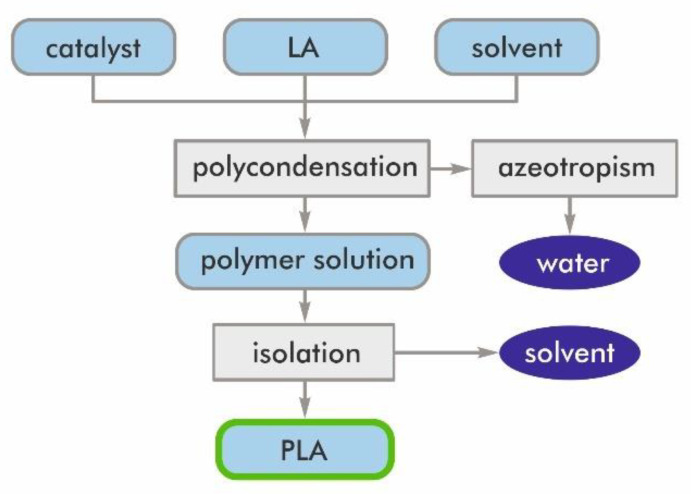
Flow diagram of the azeotropic dehydrative polycondensation of LA [28].

**Figure 8 materials-14-05254-f008:**
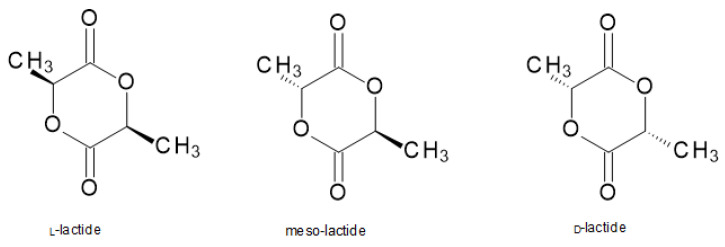
Lactide—cyclic dimers for the ROP process [7].

**Figure 9 materials-14-05254-f009:**
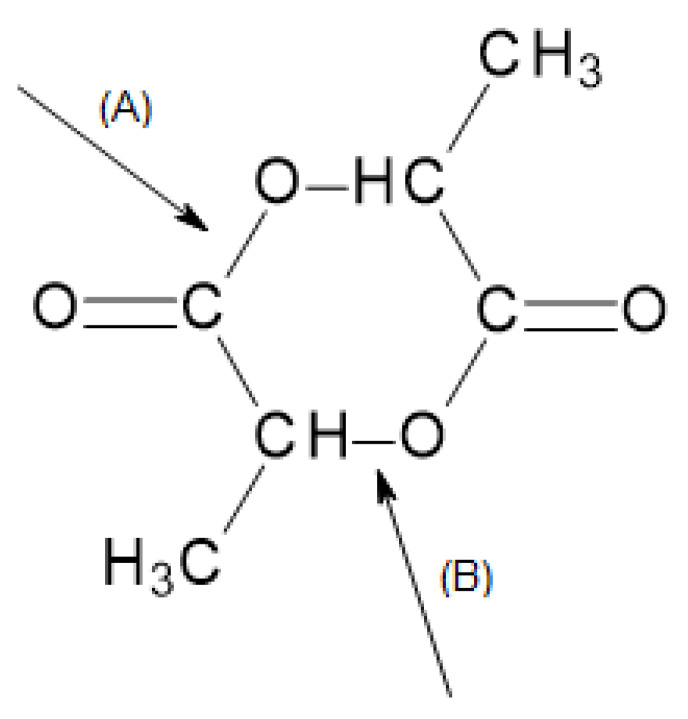
Possible positions of breaking lactide ring [5].

**Figure 10 materials-14-05254-f010:**
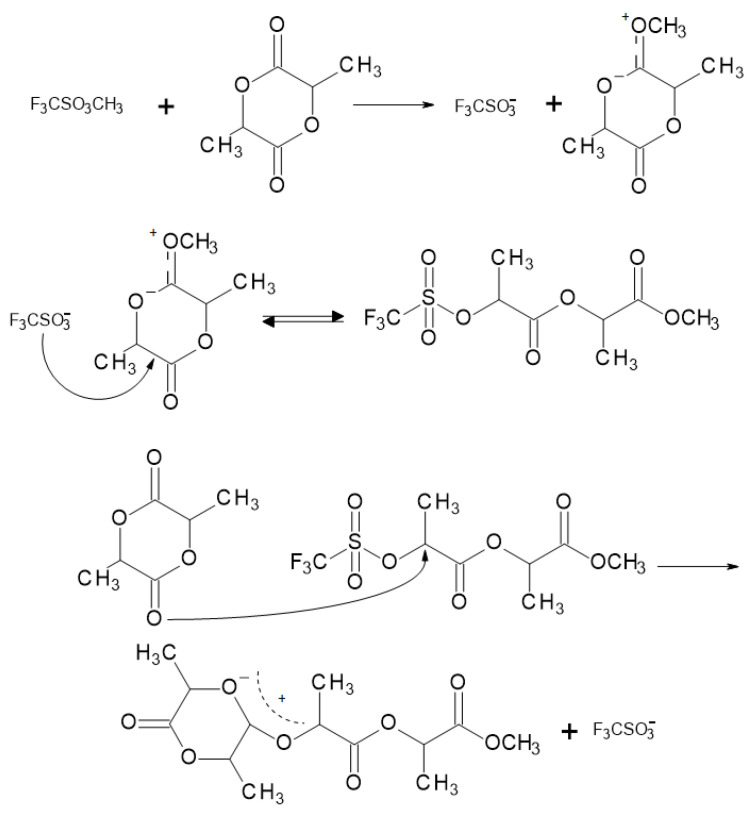
Cationic ring-opening polymerization mechanism for PLA [21].

**Figure 11 materials-14-05254-f011:**
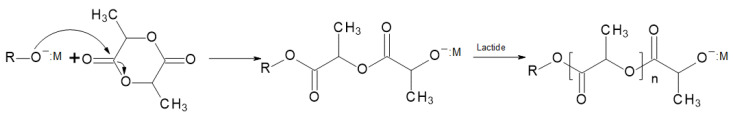
Anionic ring-opening polymerization mechanism for PLA [5].

**Figure 12 materials-14-05254-f012:**
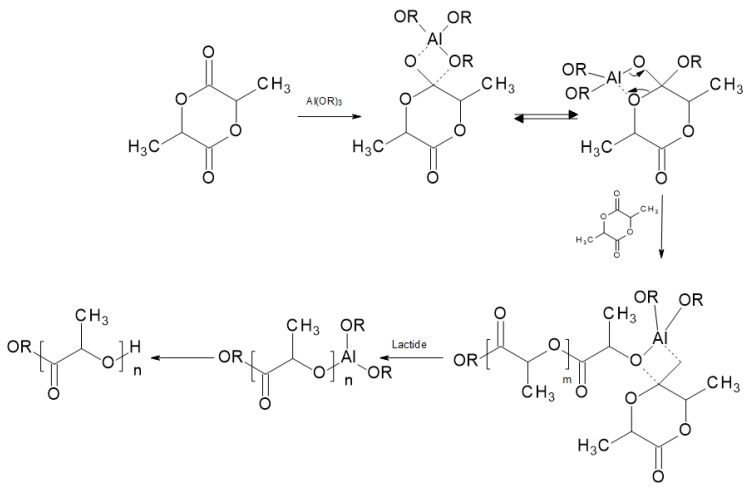
Coordinative ring-opening polymerization mechanism for PLA [5].

**Figure 13 materials-14-05254-f013:**
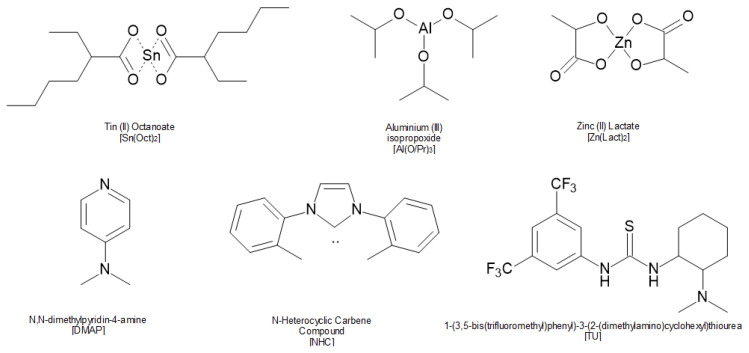
Structure of organometallic catalysts and organocatalysts used in PLA polymerization [6].

**Figure 14 materials-14-05254-f014:**
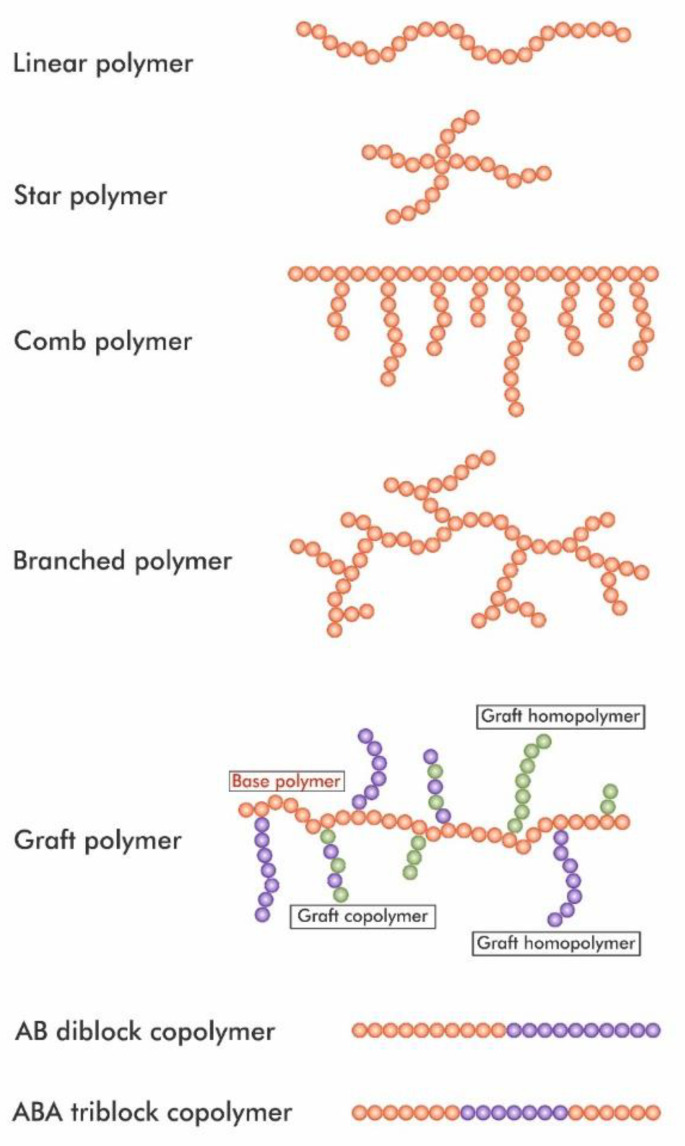
Summary of the selected polymer and copolymer architectures [50,51,52].

**Figure 15 materials-14-05254-f015:**
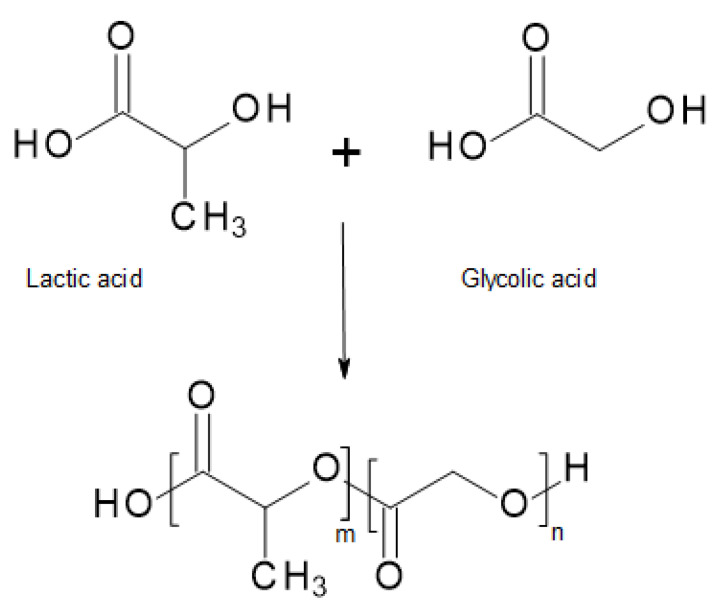
Chemical structure of LA, LG, and PLGA [53].

**Figure 16 materials-14-05254-f016:**
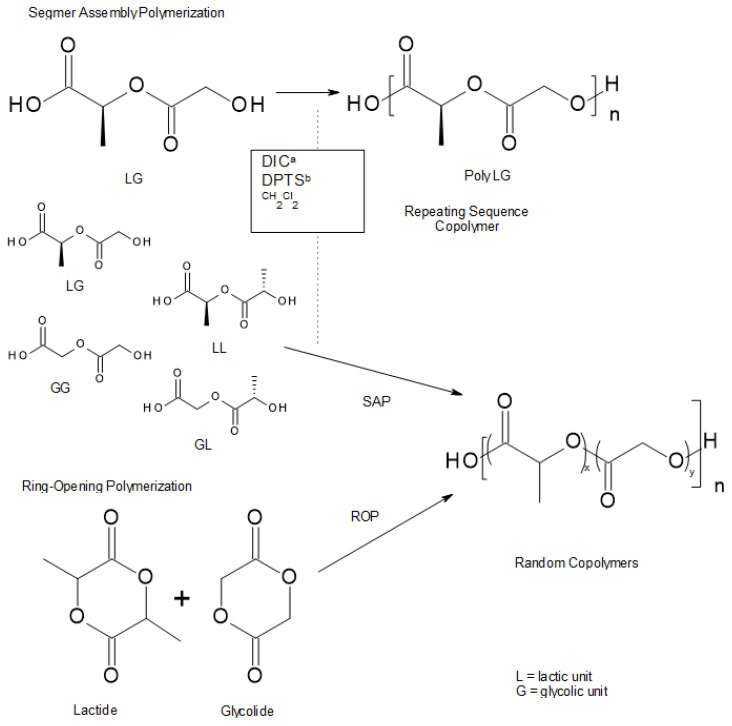
Approaches to sequenced and random PLGAs: (a) DIC = diisopropylcarbodiimide; (b) DPTS = 4-(dimethylamino) pyridinium ptoluenesulfonate. LG, LL, GG and GL are the different dimer units [58].

**Figure 17 materials-14-05254-f017:**
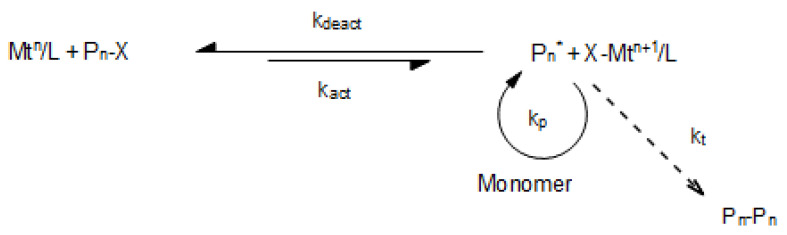
ATRP Equilibrium; k_p_—propagation reaction rate constant; k_t_—termination reaction rate constant [64].

**Figure 18 materials-14-05254-f018:**
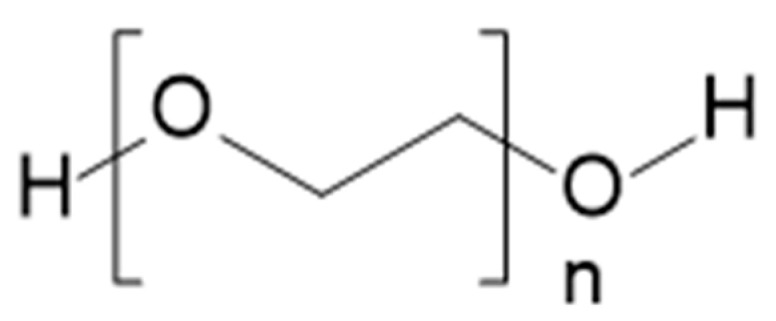
Chemical structure of PEG [66].

**Figure 19 materials-14-05254-f019:**
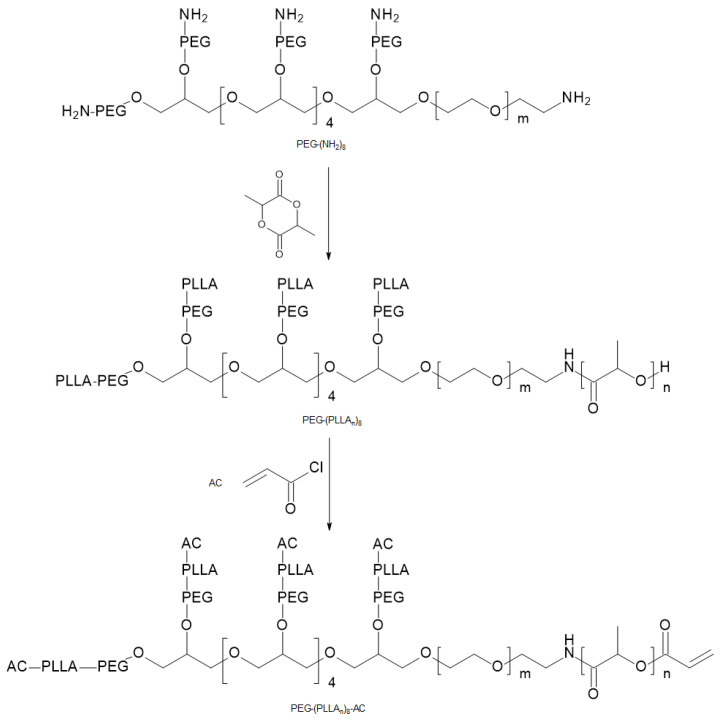
Scheme of PEG-(PLLA_n_)_8_-AC synthesis [68].

**Figure 20 materials-14-05254-f020:**
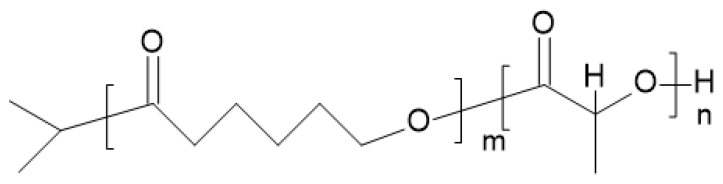
Chemical structure of diPCLPLA block copolymer [75].

**Figure 21 materials-14-05254-f021:**
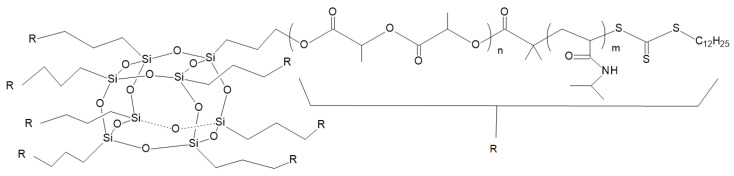
Structure of POSS(PLLA-b-PNIPAM) block copolymer [81].

**Figure 22 materials-14-05254-f022:**
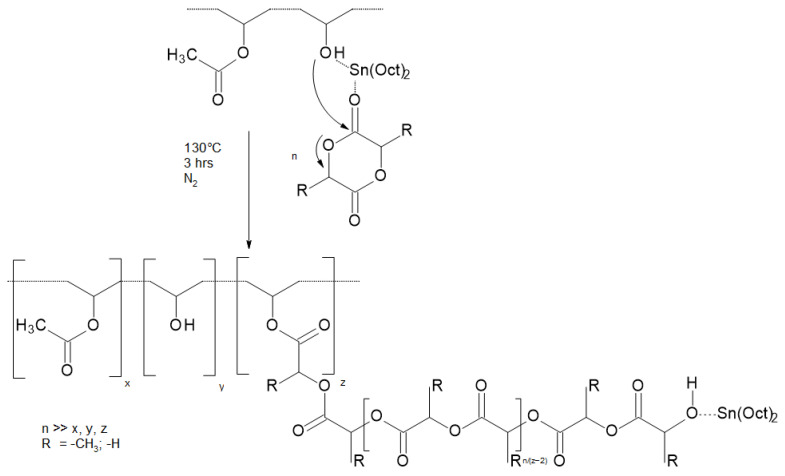
Synthesis of graft polymer PLA (PGA) on PVA by “grafting from” method [82].

**Figure 23 materials-14-05254-f023:**
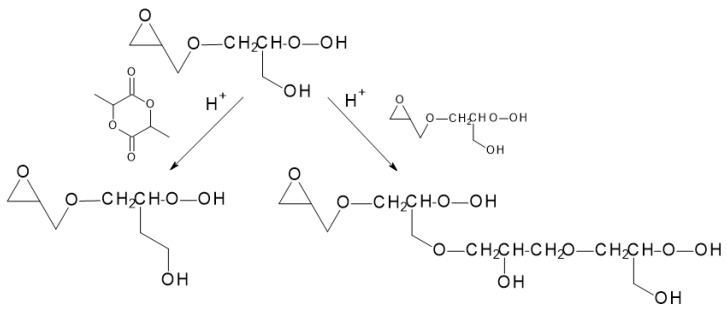
Mechanism of branched polymer formation by ring-opening copolymerization of lactide with glycidol [84].

**Figure 24 materials-14-05254-f024:**
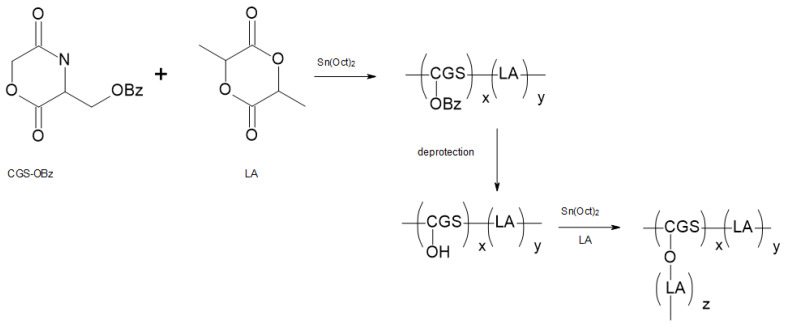
Synthesis of graft polymer PLA onto poly(LA/CGS-OBz) by the generation of initiating groups and the “grafting from” method [86].

**Figure 25 materials-14-05254-f025:**
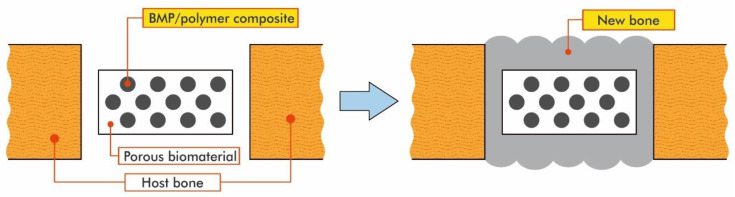
The process of repairing bone defects with the use of BMP/polymer composites [92].

**Table 1 materials-14-05254-t001:** Comparison of catalysts used in PLA synthesis.

Polymer	Catalyst	Molecular Weight	Reference
_L_-PLA	Stannous octoate	M_n_ < 250,000	[34]
_L_-PLA	Stannous octoate and compounds of titanium and zirconium	M_n_ = 40,000–100,000	[9]
_L_-PLA	Stannous octoate and triphenylamine	M_n_ = 91,000	[35]
_L_-PLA	Potassium naphthalenide	M_n_ < 16,000	[2]
_L_-PLA	Yttrium tris (2,6-di-tert butyl phenolate)	M_n_ < 25,000	[36]
_D,L_-PLA	Lanthanum isopropoxide	M_n_ = 5300–21,900	[9]

**Table 2 materials-14-05254-t002:** Degradation time of PGA, PLLA, and its copolymers.

Polymer	Degradation Time (Months) ^a^	Reference
PGA	1.5–3	[53]
PLLA	6–24	[53,54]
PLGA (LA/GA = 50/50)	1–2	[54]
PLGA (LA/GA = 50/50)	1	[55]
PLGA (LA/GA = 65/35)	1.5	[56]
PLGA (LA/GA = 75/25)	2	[56]
PLGA (LA/GA = 75/25)	4–5	[54]
PLGA (LA/GA = 75/25)	4	[57]
PLGA (LA/GA = 85/15)	5–6	[54]

^a^ Time to complete resorption.

## Data Availability

No new data were created or analyzed in this study. Data sharing is not applicable to this article.

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
