# Peer review of "Green Copolymers Based on Poly(Lactic Acid)—Short Review"

_materials, 2021, doi:10.3390/ma14185254_

Round 1
Reviewer 1 Report
Please see the document uploaded.

Author Response
Institute of Polymer and Dye Technology
Lodz University of Technology
90-924 Lodz, ul. Stefanowskiego 12/16, Poland
Tel.: +48 42 631 32 93, Fax: +48 42 636 25 43
September 03, 2021
Materials MDPI
Dear Editor,
We are resubmitting our revised paper entitled “Green copolymers based on poly(lactid acid) – short review” by Konrad Stefaniak and Anna Masek with a request to reconsider it for publication in Materials.
We have carefully considered the Editor and Reviewers' comments. The manuscript was revised exactly according to these comments. The list of responses to the reviewer’s comments and corrections made in the manuscript is attached.
For correspondence please use the following information: corresponding author: Anna Masek
Institute of Polymer and Dye Technology Technical University of Lodz
90-924 Lodz, ul. Stefanowskiego 12/16, Poland
Tel.: +48 42 631 32 13
Fax.: +48 42 636 25 43
e-mail: anna.masek@p.lodz.pl
PhD, DSc, Anna Masek
Associate Professor
Technical University of Lodz, Institute of Polymer and Dye Technology,
Stefanowskiego 12/16, 90-924 Lodz, Poland
Responses to the Reviewer's 1 comments
Main corrections in the paper are marked by green colour through the whole paper.
Reviewer 1: The manuscript reviews a synthesis of poly(lactic acid) and its copolymers, including different methods of synthesis and use of catalysts. The paper needs moderate English editing. I advise to carefully read whole paper, regarding scientific style of writing. Besides, it seems that the author did not review current works on PLA copolymerization, while most of papers cited were published after 2017. There are more recent papers on PLA copolymers for example: 10.1007/s40005-019-00442-2; 10.1016/j.polymdegradstab.2017.11.012; 10.1016/j.eurpolymj.2019.05.036; 10.1002/marc.202100100; 10.1016/j.polymer.2018.05.037; 10.3109/10837450.2015.1125920; 10.1016/j.polymer.2020.122391; 10.3390/molecules23040980; 10.1021/acssuschemeng.9b00443 and many others. Please, check the current state of the art. To keep the Journal's high scientific level this paper can be consider as proper material for publication, only after major correction of the paper.
Answer: Thank you for your positive evaluation of our manuscript and for all valuable comments that will improve the quality of the article. English has been edited. Proposed papers has been added to the manuscript as references
Reviewer 1: Abstract – line 12, which thermal properties are unwanted? Please explain.
Answer: Thank you for your comment. This fragment has been improved:
“Apart from several advantages polylactic acid has drawbacks such as brittleness or relatively high glass transition and melting temperatures.”
Reviewer 1: page 1 lines 34-35 “These reasons encourage studies on bioplastics which as a biodegradable, ecological replacement are expected to be instrumental in protection of natural environment” – it is rather limit influence on the environment than protection
Answer: We thank Reviewer for paying attention to this issue. The text has been changed:
“These reasons encourage studies on bioplastics which as a biodegradable, ecological replacement are expected to limit influence on natural environment [2].”
Reviewer 1: - page 2 lines 52-53 and line 62 „In recent years several studies were made on the subject of polylactic acid. New methods of synthesis were proposed. Several methods of PLA structural modifications were described.” – lack of the reference. Besides, the references mentioned in the manuscript are not recent there are only 11 papers 2017-2021 form total amount of 98 cited in this paper. More recent refences should be cited.
Answer: We thank Reviewer for paying attention to this issue. References have been added. Actual number of papers 2017-2021 is 17.
Reviewer 1: page 3 line 82 – “Figure 2. Scheme presenting the method of producing LA from hydrogen cyanide [5].” – LA is produced from acetic aldehyde with using of HCN.
Answer: We thank Reviewer for paying attention to this problem. The description has been improved:
“Figure 2. Scheme presenting the method of producing LA from acetic aldehyde with using of hydrogen cyanide [5].”
Reviewer 1: page 3 lines 93-93 “Polylactic acid (PLA) belongs to the family of aliphatic polyesters (Figure 4.). It is also known as polylactide, but this one has different terminal groups [5].” – please add more comment, specify the terminal groups that differ polylactide from poly(lactic acid)
Answer: We are thankful for the Reviewer’s comment. We decided to remove information about different terminal groups.
Reviewer 1: page 3 lines 98-99 – if the values for modulus is given, please add also referenced values for tensile strength and elongation
Answer: We appreciate Reviewer’s suggestions. Changes has been introduced as following:
“Poly(L-lactide) (PLLA) is marked by very good tensile strength (60 MPa), small elongation (3-4%) and high modulus (4.8 GPa) [17]. PLA is modified by adding to it plasticizers such as polyoxyethylene, polycaprolactone or citrate esters and in that PLA’s impact strength and glass transition are improved [18].”
Reviewer 1: page 4 Figure 5 – it is not clear what ratio is presented on the figure, L/D or other?
Answer: We thank Reviewer for paying attention to this issue. A following sentence has been added:
“Symbols on the legend stand for L/D-isomers ratios”
Reviewer 1: - page 5 4.1 and 4.2 sections - it is better to write first about direct polycondensation and after azeotropic dehydrative condensation
Answer: We appreciate Reviewer’s suggestions. The order has been changed.
Reviewer 1: page 5 line 122 "it was believed that a high molecular weight of PLA" - high molecular weight PLA, line 123-125 " The progress was made by Mitsui Chemicals Co., because its azeotropic dehydrative polycondensation enabled to increase the molecular weight of the polycondensation of LA. " - the molecular weight of PLA.
Answer: We are thankful for the Reviewer’s comment. Sentences have been improved as follows:
“it was believed that a high molecular weight PLA could not be prepared by the polycondensation of LA.”
“The progress was made by Mitsui Chemicals Co., because its azeotropic dehydrative polycondensation enabled to increase the molecular weight of PLA.
Reviewer 1: page 5 lines 131-133 " However, this process is currently problematic because of organic solvents usage which makes this method ecologically unattractive [19]." - the reference cited is from 2010, so "currently" is not proper
Answer: We are thankful for this comment. The sentence has been corrected as follows:
“However, this process has been problematic because of organic solvents usage which made this method ecologically unattractive [28].”
Reviewer 1: page 6 line 165 " triflate anion" - better to use systematic name
Answer: We thank Reviewer for paying attention to this issue. Systematic name has been used.
Reviewer 1: - page 7/8 lines 187-188 " They are obtained by a reaction of alcohols or phenols with ethanols of these metals" - unclear
Answer: We thank Reviewer for this comment. This sentence has been removed.
Reviewer 1: - page 10, line 243: the abbreviations LA/ML should be explained
Answer: We are thankful for this comment. The explanations has been added.
Reviewer 1: page 10, line 251: please be concise „Bi(III) acetate (Bi(OAc)3), creatinine, a Sn(Oct)2- based system and a system catalyzed by enzymes” – catalysts and a system catalyzed
Answer: We thank Reviewer for paying attention to this issue. However, in next sentences these catalysts are compared and we think that giving names of catalysts is worth it.
Reviewer 1: page 10, line 256-257 and line 263 „Catalysts systems based on aluminum alkoxides are reported to give controllable molecular weights with narrow dispersions [17].” – it should be specified the type of compound produced; please use the full name of „Al(OiPr)3” when first mentioned;
Answer: We appreciate Reviewer’s suggestions. Proposed changes has been introduced to the text:
“During the ring-opening polymerization of D-lactide, catalysts systems based on aluminum alkoxides are reported to give polylactide marked by controllable molecular weights with narrow dispersions [25]. It was found that ZnEt2 and its complex with aluminum isopropoxide (Al(OiPr)3) gave fast polymerization with low transesterification when D,L-lactide was polymerized in bulk at 150°C [46].”
Reviewer 1: page 10, line 263 „Furthermore, all the ligands are active initiating species” – what kind of ligands?
Answer: We are thankful for this comment. The sentence has been corrected as follows:
“Furthermore, all alkoxide groups of an initiator are active initiating species.”
Reviewer 1: - general: too complicated sentences like for example: page 12 lines 305-306 „Results shown in Table 1. point that adding PGA to PLLA and creating its copolymer shorten PLA degradation time even threefold from 6 to 2 months” – maybe better copolymerization of LA with GA?; page 14 lines 330-332, lines 334-337
Answer: We are grateful for drawing our attention to this problem. The sentence has been improved.
Reviewer 1: page 13 Table 1 - are there more studies on PLGA degradation?
Answer: We are thankful for this comment. New studies has been added to the Table 2.
Reviewer 1: Figure 16 – all abbreviations should be explained, what is segmer assembly polymerization?
Answer: We are grateful for this comment. “LG, LL, GG, GL” explanations has been better explained. Explanation of segmer assembly polymerization has been also added:
“Ring-opening polymerization (ROP and segmer assembly polymerization (SAP) were used. The latter is a method which allows preparing sequence polymers and gives numerous possibilities for periodic copolymer synthesis. In this approach sequenced oligomers (segmers) are first prepared and then polymerized. It is a paradigm which shows the convergence of synthetic organic and polymer chemistries [59–61].”
Reviewer 1: page 14 lines 325-326 „PGA’s positive impact on PLA degradation time should be highlighted as creating PLGA copolymer improves PLA degradation properties which is an advantage” – it was already mentioned, maybe better to describe other properties, give another conclusion
Answer: We thank Reviewer for paying attention to this issue. The conclusion has been corrected as follows:
“As suitable time of PLGA degradation is observed, this biocompatible copolymer can be used e.g. in drug delivery systems. Depending on the needs, choosing appropriate LA/GA ratio and polymer’s molecular weight allows of creating desirable material. What is more, two described methods of PLGA synthesis and wide variety of available catalysts also enable to adjust end product to the needs.”
Reviewer 1: page 15 line 346 „(a radical)” – radicals
Answer: We are thankful for this comment. The mistake has been corrected.
Reviewer 1: page 15 line 351 „Derivatives of amines and pyridines appear as ligands (L).” – unclear
Answer: We appreciate Reviewer’s suggestion. The sentence has been improved as follows:
“Derivatives of 2,2′-bipyridine such as N,N,N′,N′′,N′′-pentamethyldiethylenetriamine (PMDETA) appear as ligands (L).”
Reviewer 1: page 17 lines 416-420 – more comments
Answer: We are thankful for the Reviewer’s comment. This fragment of the text has been improved as follows:
“PCL-PLA long-chain branched block copolymer was introduced in order to prepare a biodegradable PLA material with enhanced crystallinity, rheological behavior and mechanical properties. Adding PCL-PLA copolymer to the neat PLA improved its tensile toughness without injuring abovementioned properties. Furthermore, PLA/PCL-PLA blend with 15 wt% of the PCL-PLA copolymer had much better elongation at break (210.7%) than neat PLA (7.1%). Studied copolymer was synthesized in the reaction of single hydroxyl-terminated PLA (PLA-OH) with hydroxyl-terminated 3-arm star PCL (PCL-3OH) in the presence of HMDI. HMDI was used as the chain-extending agent [73]. Toughening PLA with simultaneous preserving its biodegradability and mechanical properties should be highlighted. Subsequent studies should focus on seeking the most practical mass ratio between PLA and PCL-PLA copolymer. Described bioplastic material may find innumerous technological applications.”
Reviewer 1: - page 18 lines 430-435 – additional comments needed like for example trend for TG values, the range of 40 deg is a broad range, the author didn’t comment on two Tgs observed;
Answer: We are thankful for drawing our attention to this problem. Additional comments has been added:
“Regarding thermal properties tubular scaffolds made of poly(L-lactide-co-ε-caprolactone) (PLCL) (50:50) random copolymers were synthesized and DSC analysis showed two potential glass transition temperatures (Tg) of the scaffolds in the 0 to -40°C region. No crystalline melting peak was observed. It shows that PLCL random copolymer is amorphous. DMA profile indicated two Tg: one at -38°C and second at -11°C. This announces a phase-separated structure of studied PLCL. PCL and PLA homopolymers had higher Tg: -60°C and +50°C, respectively. Hence, the Tg at -38°C aligns with a phase composed of mainly CL unit and the Tg at -11°C signifies the other phase containing more LA moiety.”
Reviewer 1: - page 18 line 440-444 “6.5 POSS-PLA copolymer Consisting of silicon and oxygen atoms arranged in an inner eight-cornered cage with Si atoms positioned at the corners polyhedral oligomeric silsesquioxane (POSS) was used as a hybrid material [62].” – Better to write hybrid copolymers. POSS can form hybrid materials when mixing with organic species, it is not hybrid itself. There is nothing about that POSS was modified with PLA to form starshaped polyhedral oligomeric silsesquioxane multi-arm polylactides (POSS-PLA), which was further used for polyurethane formation.
Answer: We are thankful for this comment. This part of the text has been improved as follows:
“6.5. POSS-PLA hybrid copolymer
Physical and mechanical properties of PLA can be greatly improved by developing organic-inorganic hybrid materials. Consisting of silicon and oxygen atoms arranged in an inner eight-cornered cage with Si atoms positioned at the corners polyhedral oligomeric silsesquioxane (POSS) was synthesized first. Then the star-shaped POSS-polylactides (POSS-PLAs) with varied PLA arm lengths were obtained through ring opening polymerization of D,L-lactide. Eventually, the star-shaped POSS-PLA based polyurethanes (POSS-PLAUs) were formed by cross-linking POSS-PLA and polytetramethylene ether (PTMEG) with HMDI. POSS-PLAUs presented superb shape memory properties. POSS-PLAUs with shorter arm length showed faster recovery speed as a result of the higher content of POSS cores [77].”
Reviewer 1: page 18 lines 444-448 “The ring-opening polymerization of L,L-LA, catalyzed by Sn(Oct)2 was initiated by the functionalized silsesquioxane cages of the regular octahedral structure. As a result, biodegradable hybrid star-shaped POSS-PLA and linear systems with an octasilsesquioxane cage as a core and PLLA arms were given. A unique class of inorganic structures presented by POSS can be utilized in the era of hybrid polymer systems with advantageous properties [63].” The authors didn’t present any results about biodegradation of hybrids – please add proper comments.
Answer: We are thankful for the Reviewer’s comment. The explanation has been given as follows:
“Biodegradation of obtained compounds is assumed on the grounds that both lactide blocks and POSS moieties are biodegradable [78]. Biodegradable POSS lactide systems can be applied in biomedical applications.”
Reviewer 1: - page 18 lines 449-454 – POSS-g-PCL-b-PLA is mentioned but there’s nothing about that copolymerization POSS-OH initiated the ring-opening polymerization of ε-caprolactone and d,l-lactide sequentially to form the highly branched hybrid copolymer with eight PCL-b-PLA arms and further formation of hybrid copolymer-PDLLA composites was performed. Please rewrite this section.
Answer: We are thankful for this comment. This section has been rewritten as follows:
“In order to reduce the brittleness of PDLLA a highly branched hybrid copolymer based on polyhedral oligomeric silsesquioxane POSS was composed. POSS-OH was used as the core of the toughening material and then the ring-opening polymerization of ε-caprolactone and D,L-lactide was initiated sequentially to create the highly branched POSS-g-poly(ε-caprolactone)-b-poly(D,L-lactide) (POSS-g-PCL-b-PLA) copolymer with eight PCL-b-PLA arms. Furthermore POSS-g-PCL-b-PLA/PDLLA nanocomposites were prepared via solution casting method. Due to adding the PLA segment good compatibility and distribution between POSS-g-PCL-b-PLA and PDLLA matrix were observed. Elongation at break increased and the yield stress decreased as the content of POSS-g-PCL-b-PLA increased. It is caused because of the core-shell structure of POSS-g-PCL-b-PLA, which considerably improved the toughness of the PDLLA polymer matrix [80].”
Reviewer 1: - page 18-19 lines 455-466 – the formation of POSS hybrids bearing 8 chains of PLA has not been proved in the cited work [65], the mentioned hydrodynamic radius was not correlated with the self-assembly morphology of POSS(PLLA–b–PNI-PAM) block copolymer;
Answer: We are thankful for drawing our attention to this problem. This section has been rewritten as follows:
“Synthesis of the star-shaped organic/inorganic hybrid PLLA based on POSS was begun from POSS bearing octa(3-hydroxypropyl) moieties [81]. Subsequently, further transformation of POSS-PLA was made. POSS-PLA was changed into the POSS-containing star-shaped organic/inorganic hybrid amphiphilic block copolymers, poly(L-lactide)-block-poly(N-isopropylacrylamide) (POSS(PLLA-b-PNIPAM)) by the reversible addition-fragmentation transfer (RAFT) polymerization of N-isopropylacrylamide (NIPAM) (see Figure 21.). Star-shaped POSS-PLLA-b-PNIPAM amphiphilic block copolymers self-assembled into vesicles in aqueous solution. Hydrophilic PNIPAM blocks, and the hydrophobic POSS core and PLLA created coronas and the vesicular wall, respectively. The temperature dependence of the hydrodynamic radius (Rh) for POSS(PLLA12–b–PNIPAM119)8 block copolymers in aqueous solution was investigated with dynamic light scattering (DLS) measurements. When temperature decreased from 34°C to 30°C, the Rh noticeably increased from 53 nm to 93 nm. It shows the PNIPAM block in the aggregates is temperature-responsive. With temperatures below 30°C, the Rh almost did not change during the cooling or heating processes, meaning that the phase-transition process of PNIPAM block is reversible. It can be deduced that for the cooling process, with the temperature below 34°C, PNIPAM chains began to stretch. The self-assembly morphology of POSS(PLLA–b–PNIPAM) block copolymers was studied by transmission electron microscopy (TEM). Self-assembled vesicular structures of the star-shaped POSS(PLLA–b–PNIPAM) amphiphilic block copolymers in aqueous solution were observed. However, there was a broad dispersity in the size of the vesicular aggregates and the density of the vesicular wall was not uniform. The outer diameter of the vesicles was polydispersed at the range from 20 nm to 35 nm. This size of the vesicles was smaller than values measured by DLS. It results from the fact that DLS data directly reflects the size of self-assembly aggregates in solution, where the PNIPAM blocks chains are sufficiently dispersed in water, even though the PNIPAM chains are attached on the surface of the vesicular wall with their one end. Described block copolymers could be exploited in medical and biological fields [81].”
Reviewer 1: page 19 lines 469-473 “This POSS-PLA copolymer is a distinctive one among compounds described in this paper on the grounds that it is an inorganic-organic hybrid material. This copolymer desirably..” it is not clear that authors meant the POSS(PLLA–b–PNI-PAM) copolymer or POSS-PLA copolymers in general. Please rewrite to be more clear
Answer: We are thankful for this comment. This section has been corrected as follows:
“Compounds presented in this subchapter are distinctive among other examples described in this paper on the grounds that they are inorganic-organic hybrid materials. POSS-g-PCL-b-PLA/PDLLA composite desirably decreases the brittleness of a linear PDLLA which is said to be its one of the biggest drawbacks. Moreover, POSS(PLLA–b–PNIPAM) block copolymer in view of its amphiphilic and self-assembly character should be pointed out. Highlighted features of aforementioned compounds make POSS-PLA hybrid materials prospective.”
Reviewer 1: - page 19 line 475-476 ““Grafting from” method consists in e.g. grafting of PLA chains by initiating LA (or LA and GA) 475 polymerization from -OH groups of the poly(vinyl alcohol) backbone (Figure 22.)” – please correct, it is unclear; please add some comment
Answer: We are grateful for this comment. This section has been improved as follows:
“The grafting of PLA chains by initiating LA (or LA and GA) polymerization from -OH groups of the poly(vinyl alcohol) backbone is an example of the “grafting from” method (Figure 22.) [82,83]. Such action allows of modifying polymer molecular architecture. This in turn has an impact on the crystallinity and biodegradability of a polymer. Graft polymers can be applied in drug delivery systems.”
Reviewer 1: - page 20 lines 509-512 – the reference for figure 23 is not clear, please check and correct, the same for the comments
Answer: We are thankful for this comment. The reference and the comments have been corrected.
Reviewer 1: - - page 21 lines 513-516 – why better hydrophilicity is so important for PLA, please explain
Answer: We are thankful for drawing our attention to this problem. The explanation has been given as follows:
“Better hydrophilicity is important for PLA because degradation increases as material hydrophilicity increases.”
Reviewer 1: - page 21 lines 517-519 – why the melting temperature is so important for PLA synthesis, please explain
Answer: We are thankful for this comment. The explanation has been given as follows:
“The lower melting temperature occurs, the weaker forces between molecules are and as a result performing chemical synthesis is facilitated. What is more, carrying out the synthesis in lower temperature is technically more practical. When synthesis is conducted taking melting temperature into consideration also simplifies finding the purity of the obtained compound.”
Reviewer 1: - page 21 lines 534-536 – it is not clear why lower melting enthalpy thus crystallinity is important for PLA compounds
Answer: We are grateful for this comment. The explanation has been given as follows:
“Lower melting enthalpy thus crystallinity is essential for PLA compounds for the reason of degradation. Crystalline regions are more resistant to hydrolysis and as a result crystalline and semicrystalline polymers are marked by slower degradation rate than amorphous ones. [7]”
Reviewer 1: - page 21 line 537 “PLA-CGS-OBz copolymer” – maybe better to omit this abbreviation in the heading but simply write PLA copolyesters?
Answer: We are thankful for this comment. The heading has been changed.
Reviewer 1: - page 22 line 560-561 “In summary, the author’s aim of presenting PLA copolymers in this section was to show that PLA copolymerization is perspective method for improving PLA properties” – maybe better to write for manipulating/changing/ diversifying
Answer: We thank Reviewer for paying attention to this problem. The change has been made.
Reviewer 1: - page 23 lines 581-582 “However, the newly-formed bone was too small in quantity. For this reason PLA copolymers were used to solve these problems with low molecular weight PLA [20].” – unclear
Answer: We are thankful for this comment. The sentence has been corrected as follows:
“However, the newly-formed bone was too small in quantity (bone mineral density).”
Reviewer 1: - page 23 lines 583-584 “Actually, in some cases, bone defects in positions which require dynamic strength (e.g. long bone of the leg) might have to be restored.” – unclear they might have to be restored or just might be restored?
Answer: We are grateful for Reviewer’s comments. The sentence has been corrected as follows:
“Actually, in some cases, bone defects occur in positions which require dynamic strength (e.g. long bone of the leg). In these situations, in order to restore the bone, the BMP/polymer composite has to be combined with a solid biomaterial with good affinity for bone.”
Reviewer 1: - page 23 Figure 25 -the figure caption is a description, better to put this in the text
Answer: We are thankful for this comment. The change has been introduced.
Reviewer 1: - page 23 lines 603-607 – please add some comment
Answer: We are grateful for this comment. Some comment has been added:
“In vitro release mode of drug-loaded nano-particles appeared to be two-staged – a fast release in the initial stage and a slower release in a second stage. The encapsulation efficiency was 4.84% and the drug loading efficiency was 67.35%. [103]. In order to enhance abovementioned values further studies on porosity of PLGA nano-particles could be done. Changing LA/GA mass ratio in a copolymer would have potential impact on its degradation feature. Described two-staged mode of drug release can be practically applied in particular medical circumstances.”
Reviewer 1: - page 24 lines 613-617 – unclear (“It was observed that anti-tissue adhesion effect and the degradation occurred.”), please rewrite
Answer: We are thankful for this comment. The sentence has been corrected as follows:
“Another copolymer which has potentiality of being applied in medicine is PVA-g-PLA. Animal experiments featuring thin copolymer films made of this material were prepared. The thickness of the samples was 0.04-0.06 mm. In comparison with PLA homopolymer, obtained copolymer had improved hydrophilicity and flexibility. The films showed satisfying anti-tissue adhesion effect and applicable degradability in the body of the mouse. The film was entirely disappeared after 8 weeks of implantation. The films were also biocompatible, as expected, because no inflammation, hematoma or infection were noticed [107]. These results are very promising regarding future PVA-g-PLA use for preventing post-operative organ tissue adhesion.”
Reviewer 1: - page 24 lines 642-643 “It is said that PLA main drawbacks are its brittleness and thermal properties” – unclear
Answer: We are grateful for this comment. This section has been rewritten.
Reviewer 1: - page 25 lines 660-664 – there was not mention in the text about green production of PLA
Answer: We are thankful for this comment. This section has been removed.
Reviewer 1: - page 30 refs 90 and 92 – there is lack of year description for these refs
Answer: We are grateful for this comment. Mistakes have been removed.
Reviewer 1: page 14 line 336 PAA, page 15 line 338 ATRP the abbreviation should be explained at first mention; page 17 line 409 „such as PGCL and PLCL are studied”; Page 17 line 416 LB-PCL-PLA It is possible to omit this abbreviation by simply using PCL-PLA
Answer: We are grateful for Reviewer’s comments Abbreviations have been explained at first mention. LB abbreviation has been omitted.
Reviewer 1: Page 3 lines 99-101 With the aim of improving PLA impact strength and glass transition it is modified by adding to it plasticizers of various kind such as polyoxyethylene, polycaprolactone or citrate esters; Page 5 lines 130-131 The azeotropic dehydration condensation reaction of LA is a method applied to yield high molecular weight PLA; Applied to yield sounds weird, it can be written in a more simple way; Page 18 line 436-438 „Elongation….industries”; Please rewrite this part to be more proper; Page 21 lines 538-539 “Grafting from” method can be given over to prepare graft polyesters, where the polymer backbone is also a polyester. Please correct style Page 22 line 549 In order to get to know about obtained; Page 23 lines 596-570 In view of PLA’s biodegradability this polymer is also used in drug delivery systems (DDS) in which the drug can be released continuously for different periods of time up to one year.
Answer: We are grateful for Reviewer’s comments The style has been improved.
Reviewer 2 Report
Review of the manuscript “Green copolymers based on poly(lactic acid)-short review” by K. Stefaniak and A. Massek
The aim of this work is to review the current advances in PLA synthesis and copolymerization, focusing on the potential of copolymerization as a method to improve PLA properties to replace petroleum-based synthetic polymers, in particular, in pharmaceutical and biomedical applications.
The topic is interesting given the new enviromental regulations that will force the global transitions towards bioplastics. However, in the present form the paper doesn’t deserve to be accepted for publication on Materials.
In order to be considered for publication it needs some major amendments that I shown below:
- The paper needs language polishing
- Section 8 (conclusions and outlook) should be rewritten. In my opinion, there is a feeling of disorder. The ideas are barely described and without following a logical order. It is not easy to find “the predictions for future PLA syntheis, production and market development” cited in the last sentence of the abstract. For example:
- References 92-93 should be described in more detail.
- Also, in last paragraph (lines 660-664) references 97 and 98 need a wider description. The explanation given is very short in details.
- In addition, more recent references on biomedical and pharmaceutical applications of PLA copolymers, should be included in section 7 (PLA applications)
Author Response
Institute of Polymer and Dye Technology
Lodz University of Technology
90-924 Lodz, ul. Stefanowskiego 12/16, Poland
Tel.: +48 42 631 32 93, Fax: +48 42 636 25 43
September 03, 2021
Materials MDPI
Dear Editor,
We are resubmitting our revised paper entitled “Green copolymers based on poly(lactid acid) – short review” by Konrad Stefaniak and Anna Masek with a request to reconsider it for publication in Materials.
We have carefully considered the Editor and Reviewers' comments. The manuscript was revised exactly according to these comments. The list of responses to the reviewer’s comments and corrections made in the manuscript is attached.
For correspondence please use the following information: corresponding author: Anna Masek
Institute of Polymer and Dye Technology Technical University of Lodz
90-924 Lodz, ul. Stefanowskiego 12/16, Poland
Tel.: +48 42 631 32 13
Fax.: +48 42 636 25 43
e-mail: anna.masek@p.lodz.pl
PhD, DSc, Anna Masek
Associate Professor
Technical University of Lodz, Institute of Polymer and Dye Technology,
Stefanowskiego 12/16, 90-924 Lodz, Poland
Responses to the Reviewer's 2 comments
Main corrections in the paper are marked by green colour through the whole paper.
Reviewer 2: The aim of this work is to review the current advances in PLA synthesis and copolymerization, focusing on the potential of copolymerization as a method to improve PLA properties to replace petroleum-based synthetic polymers, in particular, in pharmaceutical and biomedical applications.
The topic is interesting given the new enviromental regulations that will force the global transitions towards bioplastics. However, in the present form the paper doesn’t deserve to be accepted for publication on Materials.
In order to be considered for publication it needs some major amendments that I shown below:
Answer: Thank you for your positive evaluation of our manuscript and for all valuable comments that will improve the quality of the article.
Reviewer 2: The paper needs language polishing
Answer: We are thankful for this comment. The language has been improved.
Reviewer 2: Section 8 (conclusions and outlook) should be rewritten. In my opinion, there is a feeling of disorder. The ideas are barely described and without following a logical order. It is not easy to find “the predictions for future PLA syntheis, production and market development” cited in the last sentence of the abstract. For example:
- References 92-93 should be described in more detail.
- Also, in last paragraph (lines 660-664) references 97 and 98 need a wider description. The explanation given is very short in details.
Answer: We are grateful for this comment. This section has been rewritten as follows:
“8. Conclusions
PLA is a biodegradable polymer synthesized from lactic acid. As PLA is an ecological material its use can make a positive difference on a worldwide environment.
Regarding PLA synthesis, ring-opening polymerization is a method during which lactide is converted into PLA. The reaction is catalyzed by organometal catalysts. Stannous octoate (Sn(Oct)2) is a compound commonly used as a catalyst. It gives PLA weight distributions (Mn) between 40000 and 250000. Sn(Oct)2 efficiency can be improved by adding Lewis base or distannoxane to the synthesis process. Moreover, stannous octoate can be applied in biomedical sector as it is accepted by the FDA (Food and Drug Administration, USA). However, studies concerning new catalysts that can be used in PLA synthesis are necessary as there is known negative impact of Sn(Oct)2 on PLA properties. Especially biocompatible catalysts are desirable.
Apart from many advantages of PLA such as appropriate biodegradability, durability and transparency it also has some drawbacks: brittleness, relatively high melting temperature and hydrophobicity. Hence, PLA copolymers are composed towards enhancing specific PLA properties. Diverse compounds in terms of composition (e.g. POSS-PLA hybrid copolymers) and polymer architecture (e.g. PVA-g-PLA or PLA copolyesters) are created. Due to DSC analysis graft PLA copolymers are marked by lower crystallinity and higher degradation rate than linear PLA. Sequence of individual monomers also weighs. Random PLGA degrades quicker than sequenced ones. PLA-glycidol copolymer gives significant improvement in the matter of thermal properties and better hydrophilicity. Regarding elongation at break and tensile strength features, PCL-PLA copolymers should be highlighted. It can be concluded that PLA copolymers enable creating ideal materials that can be applied in particular fields such as medicine, packaging or technology. Biocompatible PLA copolymers seem to have considerable potential in the field of biomedical applications such as drug delivery systems or tissue engineering. Studies concerning this subject should be proceeded.
Polylactic acid due to its environmentally friendly character might be a crucial material for plastics industry in the near future. Several methods of PLA synthesis and many possibilities of creating new PLA copolymers favour conducting new research. Improving PLA properties and searching for new applications of PLA seem to be the biggest challenge for the development of PLA-based materials.”
Reviewer 2: In addition, more recent references on biomedical and pharmaceutical applications of PLA copolymers, should be included in section 7 (PLA applications)
Answer: We are thankful for drawing our attention to this issue. New references have been included in section 7 (PLA applications).
Reviewer 3 Report
The paper “Green copolymers based on poly(lactid acid) – short review”, by K. Stefaniak and A. Masek, presents a revision on the synthesis, properties and applications of polylactide acid (PLA) and PLA-based copolymers.
The thematic is highly relevant to the scientific community, considering the importance of developing biodegradable and environmentally friendly polymers in order to obtain bioplastic materials with improved properties, which may find with innumerous technological applications.
However, in my opinion, the paper requires several improvements. Therefore, I only recommend its publication in Materials, after a major revision (see below).
1. English language should be improved and carefully reviewed along the manuscript.
For instance, in the conclusions (Page 24), the following sentences should be rewritten and better explained:
“Several PLA properties can be modified by different means. Experimenting with stereochemistry of PLA makes way for changing its crystallinity and glass transition temperature.”
“However, PLA copolymerization is this very method which gives considerable amount of options for modifying individual PLA features.”
“It is said that PLA main drawbacks are its brittleness and thermal properties. Examples of copolymers presented in this paper show that copolymerization is a solution for improving unwanted PLA features”
The conclusion should be more concise and should resume in what ways the use of different monomers can modify/improve the properties of PLA-based polymers.
2. The sections 1 to 4 do not present innovative information. A review paper was recently published regarding these topics: Molecules 2020, 25, 5023; doi:10.3390/molecules25215023 (which should be included in the references, as well as other recent works about PLA-based materials)
3. The section 5 (PLA synthesis – catalysts) should be improved. Discussion of possible mechanisms (catalytic cycles) for PLA synthesis using the different polymerization catalysts should be included.
4. The authors should also mention and include some bibliographic references regarding the preparation of PLA-based composites in order to improve PLA properties and applications.
5. In page 8: “The most popular catalyst used in PLA synthesis is stannous octoate - Sn(Oct)2. It is effective compound which gives high yields and high molecular weights.”
This sentence should be explained. Authors must specify the values of yields and molecular weight distributions obtained with Sn(Oct)2, and how they compare with other catalytic and non-catalytic synthetic methods
6. Section 6 – PLA copolymers. In my opinion, the topics in page 11 (lines 283-299) are related to basic concepts, so there is no need to make such explanation. Alternatively, some of this information can be embedded in the following text, with the appropriate references.
Section 6 should be improved in order to better explain the main advantages of using the different monomers to obtain the different copolymers with improved properties, comparatively with PLA.
7. Section 7 – PLA applications. This section seems incomplete. I suggest to include it in the previous section (Synthesis and applications of PLA-based copolymers).
8. The formatting of chemical structures should be uniformed, using the same template for all Figures.
Author Response
Institute of Polymer and Dye Technology
Lodz University of Technology
90-924 Lodz, ul. Stefanowskiego 12/16, Poland
Tel.: +48 42 631 32 93, Fax: +48 42 636 25 43
September 03, 2021
Materials MDPI
Dear Editor,
We are resubmitting our revised paper entitled “Green copolymers based on poly(lactid acid) – short review” by Konrad Stefaniak and Anna Masek with a request to reconsider it for publication in Materials.
We have carefully considered the Editor and Reviewers' comments. The manuscript was revised exactly according to these comments. The list of responses to the reviewer’s comments and corrections made in the manuscript is attached.
For correspondence please use the following information: corresponding author: Anna Masek
Institute of Polymer and Dye Technology Technical University of Lodz
90-924 Lodz, ul. Stefanowskiego 12/16, Poland
Tel.: +48 42 631 32 13
Fax.: +48 42 636 25 43
e-mail: anna.masek@p.lodz.pl
PhD, DSc, Anna Masek
Associate Professor
Technical University of Lodz, Institute of Polymer and Dye Technology,
Stefanowskiego 12/16, 90-924 Lodz, Poland
Reviewer 3: The paper “Green copolymers based on poly(lactid acid) – short review”, by K. Stefaniak and A. Masek, presents a revision on the synthesis, properties and applications of polylactide acid (PLA) and PLA-based copolymers.
The thematic is highly relevant to the scientific community, considering the importance of developing biodegradable and environmentally friendly polymers in order to obtain bioplastic materials with improved properties, which may find with innumerous technological applications.
However, in my opinion, the paper requires several improvements. Therefore, I only recommend its publication in Materials, after a major revision (see below).
Answer: Thank you for your positive evaluation of our manuscript and for all valuable comments that will improve the quality of the article.
Reviewer 3: 1. English language should be improved and carefully reviewed along the manuscript.
For instance, in the conclusions (Page 24), the following sentences should be rewritten and better explained:
“Several PLA properties can be modified by different means. Experimenting with stereochemistry of PLA makes way for changing its crystallinity and glass transition temperature.”
“However, PLA copolymerization is this very method which gives considerable amount of options for modifying individual PLA features.”
“It is said that PLA main drawbacks are its brittleness and thermal properties. Examples of copolymers presented in this paper show that copolymerization is a solution for improving unwanted PLA features”
The conclusion should be more concise and should resume in what ways the use of different monomers can modify/improve the properties of PLA-based polymers.
Answer: We are thankful for this comments. The language has been improved. The conclusions have been rewritten as follows:
“8. Conclusions
PLA is a biodegradable polymer synthesized from lactic acid. As PLA is an ecological material its use can make a positive difference on a worldwide environment.
Regarding PLA synthesis, ring-opening polymerization is a method during which lactide is converted into PLA. The reaction is catalyzed by organometal catalysts. Stannous octoate (Sn(Oct)2) is a compound commonly used as a catalyst. It gives PLA weight distributions (Mn) between 40000 and 250000. Sn(Oct)2 efficiency can be improved by adding Lewis base or distannoxane to the synthesis process. Moreover, stannous octoate can be applied in biomedical sector as it is accepted by the FDA (Food and Drug Administration, USA). However, studies concerning new catalysts that can be used in PLA synthesis are necessary as there is known negative impact of Sn(Oct)2 on PLA properties. Especially biocompatible catalysts are desirable.
Apart from many advantages of PLA such as appropriate biodegradability, durability and transparency it also has some drawbacks: brittleness, relatively high melting temperature and hydrophobicity. Hence, PLA copolymers are composed towards enhancing specific PLA properties. Diverse compounds in terms of composition (e.g. POSS-PLA hybrid copolymers) and polymer architecture (e.g. PVA-g-PLA or PLA copolyesters) are created. Due to DSC analysis graft PLA copolymers are marked by lower crystallinity and higher degradation rate than linear PLA. Sequence of individual monomers also weighs. Random PLGA degrades quicker than sequenced ones. PLA-glycidol copolymer gives significant improvement in the matter of thermal properties and better hydrophilicity. Regarding elongation at break and tensile strength features, PCL-PLA copolymers should be highlighted. It can be concluded that PLA copolymers enable creating ideal materials that can be applied in particular fields such as medicine, packaging or technology. Biocompatible PLA copolymers seem to have considerable potential in the field of biomedical applications such as drug delivery systems or tissue engineering. Studies concerning this subject should be proceeded.
Polylactic acid due to its environmentally friendly character might be a crucial material for plastics industry in the near future. Several methods of PLA synthesis and many possibilities of creating new PLA copolymers favour conducting new research. Improving PLA properties and searching for new applications of PLA seem to be the biggest challenge for the development of PLA-based materials.”
Reviewer 3: 2. The sections 1 to 4 do not present innovative information. A review paper was recently published regarding these topics: Molecules 2020, 25, 5023; doi:10.3390/molecules25215023 (which should be included in the references, as well as other recent works about PLA-based materials)
Answer: We thank Reviewer for paying attention to this problem. Unnecessary information has been removed from the sections 1 to 4. Molecules 2020, 25, 5023 has been added to the references. Other recent works about PLA-based materials also has been added to the paper.
Reviewer 3: 3. The section 5 (PLA synthesis – catalysts) should be improved. Discussion of possible mechanisms (catalytic cycles) for PLA synthesis using the different polymerization catalysts should be included.
Answer: We are thankful for this comment. The section 5 has been improved. Two mechanisms have been described in sections 5.1 and 5.3. Other minor corrections through the whole section 5. have been also included
Reviewer 3: 4. The authors should also mention and include some bibliographic references regarding the preparation of PLA-based composites in order to improve PLA properties and applications.
Answer: We thank Reviewer for paying attention to this issue. Information about PLA-based composites have been added to the section 3. as follows:
“Apart from PLA copolymers that are precisely described in chapter 6. various PLA-based composites are developed in order to improve PLA properties and applications.
Rodenas-Rochina et al. [22] prepared PLA/hydroxyapatite (HA) composites to apply them in devices created towards bone healing. HA micro or nanoparticles were dispersed into the polymer matrix.
Natural fibers are used in order to reinforce PLA matrix. In recent study PLA/flax, PLA/jute and PLA/falx/jute has been fabricated. The concentration of natural fibers in individual composites was varied (between 0-50%) by weight. PLA/jute and PLA/flax composites with the 40% weightage of fibers improved PLA tensile strength the most. Tensile strength of pure PLA (18.77 MPa) increased to 72 MPa and 45 MPa after flax and jute reinforcement in PLA respectively [23].
PLA/carbon fibers (PLA/CF) composites are considered to find significant applications in biomedical and engineering sectors. Excellent tensile strength and chemical stability of carbon fibers are main reasons for interest in production of PLA/CF composites [24].”
Reviewer 3: 5. In page 8: “The most popular catalyst used in PLA synthesis is stannous octoate - Sn(Oct)2. It is effective compound which gives high yields and high molecular weights.”
This sentence should be explained. Authors must specify the values of yields and molecular weight distributions obtained with Sn(Oct)2, and how they compare with other catalytic and non-catalytic synthetic methods
Answer: We are thankful for the Reviewer’s comment. The sentence has been improved as follows:
“Popular catalyst used in PLA synthesis is stannous octoate - Sn(Oct)2. It is an effective compound which gives high PLA molecular weights. Molecular weight distributions obtained with Sn(Oct)2 compared with other catalysts used in PLA synthesis are presented in Table 1.
Table 1. Comparison of catalysts used in PLA synthesis.
Polymer |
Catalyst |
Molecular weight |
Reference |
L-PLA |
Stannous octoate |
Mn < 250000 |
[34] |
L-PLA |
Stannous octoate and compounds of titanium and zirconium |
Mn = 40000-100000 |
[9] |
L-PLA |
Stannous octoate and triphenylamine |
Mn = 91000 |
[35] |
L-PLA |
Potassium naphthalenide |
Mn < 16000 |
[2] |
L-PLA |
Yttrium tris(2,6-di-tert butyl phenolate) |
Mn < 25000 |
[36] |
D-L PLA |
Lanthanum isopropoxide |
Mn = 5300-21900 |
[9] |
„
Reviewer 3: 6. Section 6 – PLA copolymers. In my opinion, the topics in page 11 (lines 283-299) are related to basic concepts, so there is no need to make such explanation. Alternatively, some of this information can be embedded in the following text, with the appropriate references.
Section 6 should be improved in order to better explain the main advantages of using the different monomers to obtain the different copolymers with improved properties, comparatively with PLA.
Answer: We thank Reviewer for paying attention to this problem. Information about polymer architecture has been embedded in the following text. Section 6 has been improved as follows:
“In summary, the aim of presenting PLA copolymers in this section was to show that PLA copolymerization is a perspective method for changing several PLA properties. Essentially, copolymerizing PLA with compounds such as PGA, PEG or glycidol shortens PLA degradation time. In authors’ judgement this is the main advantage of forming different PLA copolymers. Secondly, creating various PLA copolymers enables manipulating with PLA hydrophobicity. Mentioned ruthenium star block copolymer has PLA hydrophobic core and PAA hydrophilic corona. Besides, adding glycidol to PLA enhances PLA hydrophilicity. That has an impact on degradation rate and potential biomedical applications (e.g. drug delivery systems). Furthermore, PLA copolymers (e.g. PCL-PLA) are marked by improved toughness and/or elongation at break in comparison with neat PLA. Depending on the needs, diversifying other PLA properties such as crystallization or contact angle values makes possibilities of composing new PLA copolymers. The fact that both inorganic (e.g. POSS) and organic compounds can be used for PLA copolymerization is worthy of noticing. Many possibilities concerning designing different copolymer architectures also aid this method’s advance. As a result, PLA copolymers with broad applications can be obtained – next chapter is focused on this issue.”
Reviewer 3: 7. Section 7 – PLA applications. This section seems incomplete. I suggest to include it in the previous section (Synthesis and applications of PLA-based copolymers).
Answer: We are grateful for this comment. Section 7 has been expanded and new references have been added.
Reviewer 3: 8. The formatting of chemical structures should be uniformed, using the same template for all Figures.
Answer: We are thankful for this comment. All chemical structures have been made in the same way. Figures 3, 6, 16, 22 have been improved.
Round 2
Reviewer 1 Report
Thanks for the detailed corrections. I accept the manuscript in the present form.
Reviewer 2 Report
In my opinion the corrections made by authors, following the reviewers suggestions, have greatly improved the quality of the manuscript and it's worthy of publication .
Reviewer 3 Report
The authors have addressed all my comments, and have performed the suggested modifications and the required improvements. So, in my opinion, the paper can now be accepted for publication in Materials journal.